# Maintenance of quiescent oocytes by noradrenergic signals

Jeongho Kim[1,6], Moonjung Hyun [2,4,6], Masahiko Hibi [3] & Young-Jai You[2,3,5✉]

All females adopt an evolutionary conserved reproduction strategy; under unfavorable conditions such as scarcity of food or mates, oocytes remain quiescent. However, the signals to maintain oocyte quiescence are largely unknown. Here, we report that in four different species – *Caenorhabditis elegans*, *Caenorhabditis remanei*, *Drosophila melanogaster*, and *Danio rerio* – octopamine and norepinephrine play an essential role in maintaining oocyte quiescence. In the absence of mates, the oocytes of *Caenorhabditis* mutants lacking octopamine signaling fail to remain quiescent, but continue to divide and become polyploid. Upon starvation, the egg chambers of *D. melanogaster* mutants lacking octopamine signaling fail to remain at the previtellogenic stage, but grow to full-grown egg chambers. Upon starvation, *D. rerio* lacking norepinephrine fails to maintain a quiescent primordial follicle and activates an excessive number of primordial follicles. Our study reveals an evolutionarily conserved function of the noradrenergic signal in maintaining quiescent oocytes.

[1] Department of Biological Sciences, Inha University, Incheon 22212, South Korea. [2] Department of Biochemistry and Molecular Biology, Virginia Commonwealth University, Richmond, VA 23298, USA. [3] Graduate School of Science, Nagoya University, Nagoya 464-8602, Japan. [4]Present address: Biological Resources Research Group, Bioenvironmental Science & Toxicology Division, Korea Institute of Toxicology (KIT), Gyeongsangnam-do 52834, South Korea. [5]Present address: Department of Internal Medicine, University of Texas Southwestern Medical Center, Dallas, TX 75390, USA. [6]These authors contributed equally: Jeongho Kim, Moonjung Hyun. ✉email: Young-Jai.You@UTsouthwestern.edu

Sexual reproduction in animals requires timely fusion of an egg and a sperm; the right environment at the right timing in producing and releasing gametes are essential. While in spermatogenesis, meiosis and differentiation proceed continuously without cell cycle arrest, in oogenesis, meiosis is arrested and reinitiated upon signals in order to not waste eggs, which contain huge amounts of resources.

In mammals, the primordial oocytes produced at birth remain quiescent, arresting at diplotene of the meiotic prophase I, and can remain quiescent for as long as their entire reproductive span. At puberty, upon release of the gonadotropins follicle stimulating hormone (FSH) and luteinizing hormone (LH), primordial follicles consisting of a meiotically arrested oocyte and surrounding granulosa cells undergo activation (termed PFA: primordial follicle activation), resume growth, and the oocytes complete meiosis I to fully mature.

The timing of oocyte awakening is the key to reproduction, not only by providing available oocytes with the right timing, but also by preserving the pool of available and high-quality oocytes. Indeed, the process is tightly regulated by coordinated mechanisms involving timely release of endocrine hormones, constant communications among granulosa cells and the oocyte via gap junctions and juxtacrine signals, and assessment of the nutritional state of the female. In mammals, a signal of natriuretic peptide precursor type C (NPPC) from granulosa cells is necessary for maintaining high cAMP concentration in oocytes, which is required to maintain meiotic arrest[1,2]. LH whose receptors are expressed only after secondary follicles grow to small antral follicles releases the oocyte from meiotic arrest by antagonizing the NPPC signal in granulosa cells[3]. The mTORC1 signaling, critical in linking the nutritional state of an animal to its cellular metabolism, plays an essential role in awakening quiescent oocytes. KIT ligand (KITL) secreted from the pre-granulosa cells upon mTORC1 activation binds to the KIT receptors in the primordial oocyte, which activates the PI3K-AKT pathway in the oocyte and results in cytoplasmic sequestration of FOXO3A, leading to PFA[4]. Reduced mTOR signaling prevents quiescent oocytes from awakening, whereas excessive mTOR signaling awakens all quiescent oocytes prematurely[4].

The signals to maintain the quiescent primordial follicles, however, are still not fully understood. Anti-Müllerian hormone (AMH), a member of the TGFβ superfamily produced by granulosa cells, inhibits the PFA[5] and FSH-mediated steroidogenesis[6]. Female $Amh^{-/-}$ mice carry a smaller number of primordial follicles as they age compared to the control, despite a normal appearance of the ovary and litter size[7], suggesting loss of primordial follicles due to excess activation. The level of AMH is temporarily increased immediately after a PFA then falls, indicating that locally acting AMH prevents the upcoming PFA, limits the number of recruited primordial follicles for activation at a given time and maintains the quiescent oocyte reserve. However, when oocytes are released from the follicles into a culture medium in vitro, oocytes are spontaneously released from quiescence[8]. Also, when fetal ovaries were severed from their nerves, isolated and cultured in a serum-free medium, the oocytes successfully underwent PFA in vitro[9,10]. These observations suggest the existence of signals besides AMH that are necessary for maintaining quiescence and inhibitory to PFA originated from outside of the ovary.

Zebrafish, a widely used small animal model, spawn once a year only during a monsoon in the wild[11], indicating that the activation and the growth of primordial follicles is tied to availability of food. The growing oocytes are divided into five stages of primary growth (stage (I), cortical alveolus stage (II), vitellogenesis (III), maturation (IV), and mature egg (V)) based on size and morphology[12]. Quiescent oocytes arrested in meiosis prophase I

are maintained at stage I. As in mammals, once awakened, the oocytes grow to stage IV by actions of FSH and LH[12]. The oocyte growth and maturation in zebrafish and mammals share several molecular mechanisms; the KITLs secreted from the granulosa cells act on the receptors residing in the oocytes, potentially linking nutrients signals such as insulin to the growth of oocytes. Although AMH does not play the same role in the fish as in mammals, other TGFβ family members such as inhibin, activin, and BMP15 play important roles in inhibiting and promoting the oocyte maturation mostly from stage II − IV by controlling the sensitivity to gonadotropins, as AMH does[13]. The signals to maintain quiescent oocytes, however, have not been identified.

The two most studied invertebrate models, D. melanogaster and C. elegans, also maintain quiescent oocytes. In D. melanogaster, oocyte quiescence can be induced by short photoperiod conditions at low temperature or starvation[14]. Flies at low temperature rarely feed, suggesting that starvation alone could be a sufficient cause of ovarian quiescence[15]. The fat body serves similar functions to those of the adipose tissue and the liver in vertebrates and regulates reproduction by nutritional status[16]. Under well-fed conditions, the fat body secrets Stunted (Sun), which binds to its receptor Methuselah (Mth) in the insulin producing cells (IPCs), which in turn promotes secretion of Drosophila Insulin-Like Peptides (dILPs) from the IPCs[17]. The high insulin signal produced in response to a good nutritional state promotes reproduction by (a) increasing the rate of cell division in the germline stem cells[18] and by (b) increasing the level of juvenile hormone (JH), whose activity is absolutely required for vitellogenesis beginning from the stage 8 egg chamber to accumulate yolk[19]. Upon binding to its receptors, Germ-cell-expressed (Gce) and Methoprene-tolerant (Met), JH upregulates the expression of vitellogenin receptors as well as the genes involved in vitellogenin synthesis[20]. When the fly is starved, the ovaries become quiescent and the levels of insulin and JH are reduced[21,22]. However, whether reduction of insulin and JH is sufficient to maintain quiescent ovaries, or another inhibitory signal is necessary remains unknown.

C. elegans hermaphrodites are self-fertilizing and each produces ~300 sperm before the onset of oogenesis. C. elegans oogenesis occurs inside the gonadal sheath, which is functionally equivalent to somatic follicle cells in vertebrates and flies. Competent oocytes at the diakinesis stage (hereafter called diplotene stage) in meiosis are self-fertilized by sperm stored in the spermatheca at a constant rate. As the sperm are made and stored before oogenesis and are ready to fertilize as soon as an oocyte is ready, the second meiotic arrest at the metaphase present in most gonochoristic animals does not exist in C. elegans. The most proximal oocyte is activated by the sperm signal, and an oocyte matures every 23 min under well-fed conditions[23]. When sperm are exhausted, quiescent oocytes accumulate inside the gonadal sheath. The signal required to maintain quiescent oocytes in the absence of sperm has not been identified.

The adrenergic innervation of the ovary is thought to be required for modulation of the function of ovarian vasculature and ovarian sheath cells. Despite the reports from multiple animals that the ovaries are innervated by cells expressing norepinephrine in vertebrates or octopamine in invertebrates, the function of noradrenergic innervation of the ovary is unclear. In D. melanogaster, octopamine is required for the egg-laying process[24]. However, C. elegans mutants lacking octopamine, and zebrafish and mouse mutants lacking norepinephrine reproduce normally[25–27], indicating that the role of noradrenergic signaling in fertility might be hidden under well-fed laboratory conditions. Because of the well-known roles of noradrenergic signals in the responses to stresses such as the fight-or-flight response[28] and to starvation[29], we investigated the function of norepinephrine and

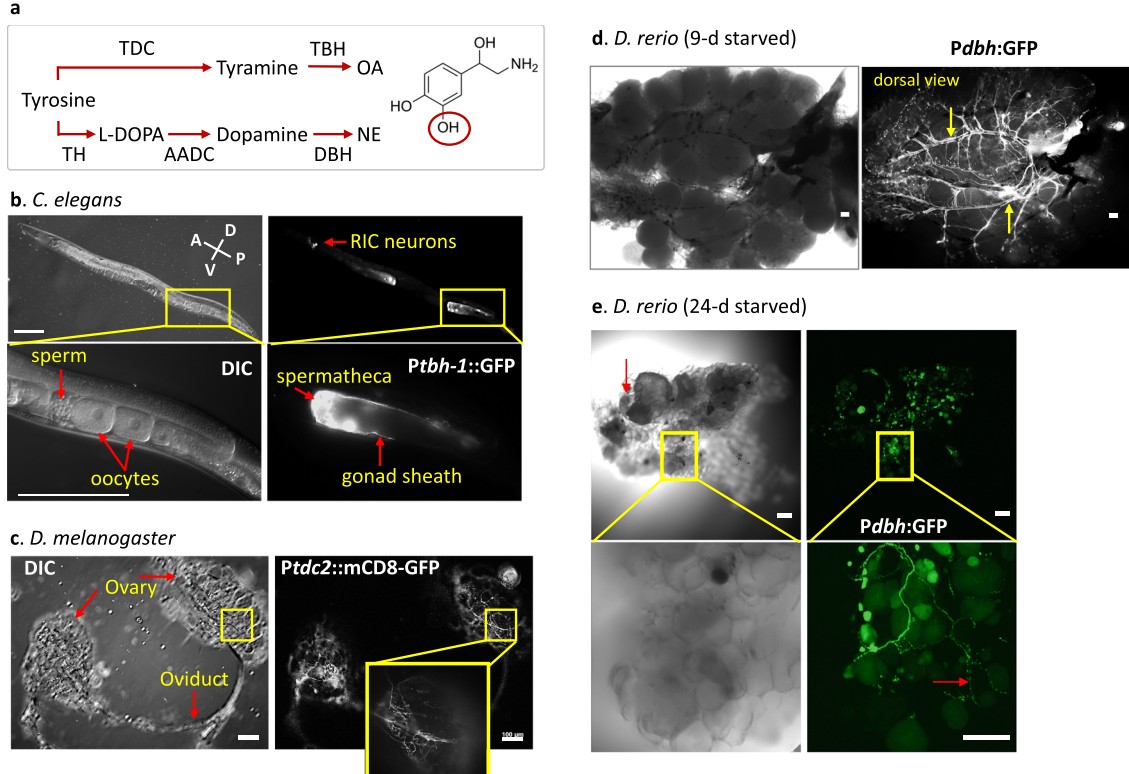

**Fig. 1 Noradrenergic innervation of ovaries in the three animals. a** The synthesis pathways of norepinephrine (NE) and octopamine (OA) and the enzymes for each step. TDC: Tyrosine decarboxylase, TBH: Tyramine β-hydroxylase, TH: Tyrosine hydroxylase, AADC: Aromatic L-amino acid decarboxylase, DBH: Dopamine β-hydroxylase, L-DOPA: L-Dihydroxyphenylalanine. NE is shown. The hydroxyl group (red circle) is absent in OA. **b** GFP driven by a *tbh-1* promoter ($P_{tbh-1}$::GFP) is expressed in the gonadal sheath cells and the spermatheca in addition to RIC interneurons in the head of a 1-day old adult *C. elegans*. All 15 *C. elegans* examined showed similar expression pattern. The left panel shows developing oocytes surrounded by gonadal sheath (top) and the enlarged image of the inset in a DIC image (bottom). The right panel shows the GFP expression of *tbh-1*. The highest expression is observed in the spermatheca and the proximal three pairs of the gonadal sheath cells that surround diakinesis-staged oocytes. GFP is also expressed in a pair of RIC neurons in the head. **c** The expression of a *tdc2-GAL4* reporter with a membrane targeting mCD8-GFP marks OA-expressing ovarian neurons originated from the abdominal nerves in a virgin female *D. melanogaster*. The neuronal branches cover the entire surface of the ovary. Similar *tdc2*-expression was observed from 18 flies. **d** EGFP driven by a *dbh* promoter is expressed in *dbh*-expressing neurons that surround the entire surface of the ovary in *D. rerio*. An image from a light microscopy shows many mature and growing oocytes in the ovary of 9-day starved zebrafish. The *dbh*-expressing nerves run along the main vasculatures on the dorsal side of the ovary (arrows). Similar *dbh*-expression was observed from 8 fish. **e** DIC and confocal images of the ovary of a 23-d-starved zebrafish. Degenerating oocytes are shown (red arrow). A higher magnification image (bottom) shows small-sized varicosities along the neurons (red arrow). Similar *dbh*-expression at lower magnification was observed from four fish. **b**–**e** Scale bar = 100 μm.

octopamine in ovaries under unfavorable conditions, where the maintenance signal for oocyte quiescence will be manifest. Here, we show that noradrenergic signal is essential to maintain quiescent oocyte pools in all four model organisms of *C. elegans*, *C. remanei*, *Drosophila* and zebrafish under the stressful conditions, revealing a new role of noradrenergic signaling in safeguarding oocytes.

## Results

**Noradrenergic innervation of ovaries in the three animals.** Norepinephrine (NE) and its invertebrate counterpart octopamine (OA) are produced by tyrosine hydroxylase (TH) + dopamine β hydroxylase (DBH), and by tyrosine decarboxylase (TDC) + tyramine β hydroxylase (TBH), respectively (Fig. 1a). Using transgenic animals carrying green fluorescent protein (GFP) fused with promoters of *tbh-1* in *C. elegans*, *tdc2* in *D. melanogaster*, and *dbh* in *D. rerio*, we observed that in all cases, the ovaries were heavily innervated by processes that express the enzymes (Fig. 1b–e). This suggests that in all three species, NE or OA are synthesized and released to the ovary.

In *C. elegans*, consistent with a previous report[25], OA synthetic enzymes are highly expressed in the spermatheca and the

proximal gonadal sheath (Fig. 1b). *tbh-1* expression in the gonadal sheath begins from the late larval stage 4 (L4), when diplotene-stage oocytes appear, while *tbh-1* expression in RIC interneurons in the head begins as early as at the L1 stage (RIC is indicated in Fig. 1b). This *tbh-1* expression in gonadal sheath cells accounts for the observation that adult *C. elegans* produce at least five times more OA than larvae, which express OA only in RIC[29]. The level of TBH-1 is the strongest in the proximal three pairs of gonadal sheath cells, which cover the meiotic oocytes in-waiting, suggesting a potential OA function in oocyte quiescence. Strong expression in the spermatheca (see Fig. 2a, Sp.), which are involved in fertilizing and moving the fertilized eggs, suggests a role of OA in ovulation, as has been reported in *D. melanogaster*[30].

In *D. melanogaster*, each ovary consists of ~15 sacs of ovarioles, which contain a series of developing egg chambers in an assembly-line fashion. Oogenesis begins in the germarium where germline stem cells reside. The stage-2 egg chambers emerge from the germarium surrounded by a layer of somatic follicle cells. The egg chamber rapidly develops into the full-grown stage-14 oocyte under well-fed conditions. Each ovariole is surrounded by a single layer of contractile epithelium, the epithelial sheath, and then

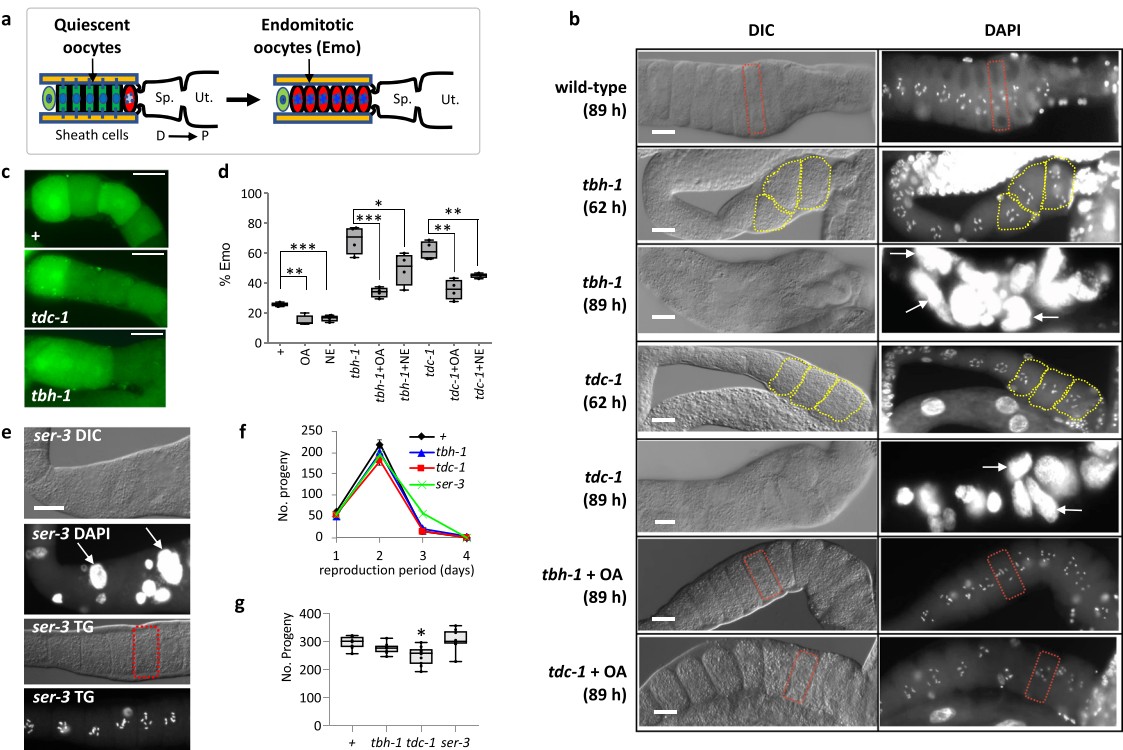

**Fig. 2 Maintenance of quiescent oocytes by octopamine in *C. elegans*. a** Progress to Emo after sperm depletion (quiescent oocytes in green, Emo in red, the gap junctions between oocytes and sheath cells in blue). D: distal, P: proximal, Sp.: Spermatheca, Ut.: Uterus. **b** Representative images of DIC (left) and DAPI staining (right) of oocytes (rectangles: quiescent oocytes, polygons: oocytes, arrows: Emo). The images represent 84 gonad images of wild-type, 9 gonads of 62 h of *tbh-1*, 5 gonads of *tdc-1*, 94 gonads of 89 h of *tbh-1*, 77 gonads of *tdc-1*, 66 gonads of *tbh-1* + OA, 66 gonads of *tdc-1* + OA. **c** Representative images of antibody staining of activated MAPK in the proximal oocytes[42]. Staining frequency: 77% for wild-type (*n* = 35 gonads), 65% for *tdc-1* (*n* = 17 gonads), and 70% for *tbh-1* (*n* = 30 gonads). **d** Quantification of Emo at 89 h after hatch. Two-sided *t*-test: *p* = 0.001 (wild-type (*n* = 113 animals) vs. wild-type + OA (*n* = 99 animals)), 0.0003 (wild-type vs. wild-type + NE (*n* = 109 animals)), 0.0004 (*tbh-1* (*n* = 132 animals) vs. *tbh-1* + OA (*n* = 99 animals)), 0.035 (*tbh-1* vs. *tbh-1* + NE (*n* = 111 animals)), 0.001 (*tdc-1* (*n* = 129 animals) vs. *tdc-1* + OA (*n* = 103 animals)), 0.002 (*tdc-1* vs. *tdc-1* + NE (*n* = 86 animals)) (\**p* < 0.05, \*\* *p* < 0.005, \*\*\**p* < 0.001). **e** A *ser-3* mutant shows Emo. Transgenic lines (*ser-3* TG, *n* = 14 gonads) rescue *ser-3* mutants (*n* = 31 gonads). **f, g** The reproduction periods (**f**) and the total brood sizes calculated from **f** (**g**) of *tbh-1* (*n* = 8 animals, two-sided *t*-test: *p* = 0.059), *tdc-1* (*n* = 10 animals, *p* = 0.003) and *ser-3* mutants (*n* = 9 animals, *p* = 0.623). *p*-value is vs. wild type. The mutants and wild-type (*n* = 8 animals) stop reproducing after 3 days as adults. Boxplots show the median, mean (X), interquartile range (IQR). The upper whisker: the maxima smaller than 1.5 times IQR plus the third quartile, the lower whisker: the minima larger than 1.5 times IQR minus the first quartile. **b, c, e** Scale bar = 20 μm. Source data are provided as a Source Data file.

collectively enclosed by the contractile peritoneal sheath that surrounds each ovary. Both sheaths surround egg chambers in a mesh-like fashion so that all egg chambers are accessible to OA in the hemolymph[31]. A GAL4 line that drives GFP expression by a *tdc2* promoter showed that *tdc2* is expressed in all processes and many varicosities or boutons of neurons covering the ovary (Fig. 1c)[24].

In *D. rerio*, the ovary is composed of ovigerous lamellae[12], each of which contains developing follicles where oocytes of different developmental stages reside. The GFP expression driven by a *dbh* promoter shows that the ovary is innervated by norepinephrinergic neurons (Fig. 1d). The nerves run along the main vasculatures of the dorsal side of the ovary. In the cortical areas, the nerves project extensively across the entire lamella, seemingly independent of the vasculature (Fig. 1e, insets)[32]. The processes labeled with GFP consist of many small-size varicosities or boutons from which NE would be released to the ovary, although we did not observe direct innervation of the follicles by *dbh*-expressing neurons. Taken together, these observations suggest that NE is synthesized and released into the zebrafish ovary.

**Maintenance of quiescent oocytes by OA in *C. elegans*.** Due to lack of the second meiotic arrest, when kept unfertilized for long periods of time, a small fraction of *C. elegans* oocytes awaken, spontaneously undergo several rounds of abnormal haploidic mitotic replications and become polyploid endomitotic oocytes. This is called the Endomitotic Oocytes (Emo) phenotype[33] (Fig. 2a). While the first oocyte (red) is exiting quiescence and resumes meiosis, the remaining oocytes remain quiescent. If the arrest signal is absent and/or the most proximal oocyte blocks the gonad-spermatheca passage, however, the rest of the oocytes become Emo.

After ~90 h from hatching, sperm are depleted and 25% of wild-type *C. elegans* exhibit Emo, whereas 80% of the *tbh-1* mutants exhibit Emo (Fig. 2b, d). The *tbh-1* mutants carry a deletion covering the 5th and 6th exons and do not produce OA[25]. In the presence of sperm, the overall reproduction processes of oocyte maturation (examined by MAPK activation), ovulation, and fertilization of *tbh-1* and *tdc-1* mutants are indistinguishable from those of wild-type, showing that OA is not required for oocyte maturation or any process beyond (Fig. 2c and Supplementary videos 1–3). Also, the mutants did not deplete their sperm any sooner than wild-type animals, as the mutants and wild-type stop producing progeny after 3 days as adults. This shows the mutants' reproduction span is similar to that of wild type (Fig. 2f, g), although the *tdc-1* mutation reduces brood size

slightly. The Emo phenotype is rescued by adding OA (20 mM) exogenously to the mutants (Fig. 2b, d), indicating that OA is required for maintaining oocyte quiescence and that the Emo phenotype is the result of failure to maintain quiescence. Interestingly, when we treated *tdc-1* or *tbh-1* mutants with NE (5 mM), it partially rescued the Emo phenotype of the mutants, showing that NE could at least partially replace OA function in oocyte quiescence in *C. elegans* (Fig. 2d).

To identify the OA receptor for oocyte quiescence, we performed RNAi of all five reported OA receptors, *octr-1*, *ser-3*, *ser-6*, *tyra-2*, and *tyra-3*[34] and found that only *ser-3* RNAi produced the Emo phenotype. The *ser-3* mutants phenocopied the RNAi result, confirming that SER-3 is the OA receptor (Fig. 2e). It has been suggested that the quiescence signals are produced in gonadal sheath cells that surround oocytes and are then transported to oocytes through gap junctions[35]. SER-3 expression is limited to head muscles, a few neurons, intestine, spermatheca, and gonadal sheath cells[36]. To identify the SER-3 site of action, we targeted *ser-3* expression to gonadal sheath cells using a *ceh-18* promoter. CEH-18 is a Pit-1/Oct-1,2/Unc-86 (POU) domain-containing transcription factor required for gonadal sheath cell differentiation[37]. Although CEH-18 is broadly expressed in muscles, neurons, and the gonads[38], within the gonad it is only expressed in gonadal sheath cells and not detected in sperm or oocytes[39]. In addition, the only tissues where both *ceh-18* and *ser-3* are expressed are gonadal sheath cells and spermatheca. This construct rescues the Emo phenotype in *ser-3* mutants (Fig. 2e, *ser-3* TG), suggesting that SER-3 function in gonadal sheath cells is sufficient to maintain quiescent oocytes.

**Maintenance of quiescent oocytes by OA in female nematodes**. Next, we tested whether lack of OA results in the Emo phenotype in feminized mutants of *C. elegans* and in females of a related species, *C. remanei*. When we knocked down *tdc-1* expression by RNAi in two feminized mutants of *C. elegans*, *fem-1* and *fog-2*[40,41], the mutants failed to maintain oocyte quiescence and the oocytes became Emo beginning at 24 h after the 4th larval stage (L4), the last larval stage before the animal becomes an adult (Fig. 3a). When we knocked down *tdc-1* gene expression by RNAi in *C. remanei*, a gonochoric (i.e., having males and females) species, the oocytes of the virgin became Emo beginning 24 h after the L4 stage (Fig. 3b). Together, our results suggest that OA serves as an oocyte quiescence signal in *C. elegans* and *C. remanei*.

Oocyte maturation is initiated by sperm signal Major Sperm Proteins (MSPs) and mediated via GSA-1, a canonical $G_{\alpha s}$ that activates an adenylyl cyclase to produce cAMP and activate protein kinase A (PKA) in gonadal sheath cells[35,42]. Any mutations in this process block the oocyte maturation and delay ovulation even in the presence of sperm. The oocytes of OA mutants mature normally when the sperm is present, indicating the oocyte maturation process triggered by sperm signal is intact. Yet, because OA maintains oocyte quiescence when sperm is absent, once sperm is present, sperm signal would override OA signaling. To examine the genetic interaction between OA and sperm signaling, we first generated *fog-2; tbh-1* double mutants, which were maintained in the presence of males. Consistent with the RNAi results, *fog-2; tbh-1* double mutants failed to maintain quiescent oocytes from as young as 10 h after L4, and the oocytes became round instead of remaining cylinder-shaped as in *fog-2* single mutants (Fig. 3c). *fog-2; tbh-1* double mutants laid oocytes at a higher ovulation rate than *fog-2* single mutants (Fig. 3d) and became Emo at 1 d (19% Emo, *n* = 31), 2 d (50% Emo, *n* = 24), and 3 d (67% Emo, *n* = 24) after L4. In contrast, *fog-2* females maintain quiescent oocytes and produce no Emo for 3 days (0% Emo, *n* = 35). Exogenous OA rescued *fog-2; tbh-1* both for

ovulation rate (albeit in a delayed manner, likely because pharmacological action of exogenous OA action takes time) and Emo as quiescent oocytes accumulated (Fig. 3c, d) (0% Emo, *n* = 24).

Next, we examined the genetic interaction between OA and sperm signaling by comparing the ovulation rates between *fog-2; tbh-1* and *fog-2* under the condition of reduced sperm signaling of $G_{\alpha s}$, which is encoded by *gsa-1*. We reasoned that if OA directly interacts with sperm signaling, the *gsa-1* phenotype would be epistatic to that of *tbh-1* in maintaining quiescent oocytes; the phenotype of *gsa-1*RNAi would be the same as that of *gsa-1*RNAi in the *tbh-1* background. As $G_{\alpha s}$ is also required for spermatheca contraction[43], the oocytes of *gsa-1* RNAi-treated *fog-2; tbh-1* animals became trapped in the spermatheca, where they matured and became Emo in the gonadal sheath, whereas the oocytes of *fog-2* single mutant did not (Fig. 4a, b). Although it is unclear why oocytes of *fog-2* mutants were able to pass spermatheca in *gsa-1* RNAi treatment, the fact that *gsa-1* RNAi produces different phenotypes between *fog-2* and *fog-2; tbh-1* indicates *gsa-1* is not epistatic to *tbh-1*.

As *gsa-1* RNAi-treated *fog-2; tbh-1* mutants did not lay unfertilized oocytes, we could not accurately determine the ovulation rate. Instead, we monitored the entire progress of ovulation of a single live animal. Approximately 36 h after the L4 stage (*t* = 0 in Fig. 4a, b), 100% of *gsa-1* RNAi-treated *fog-2* mutants contained 12 or more cylinder-shaped quiescent oocytes in each gonad arm. In contrast, only 17.7% of *gsa-1* RNAi-treated *fog-2; tbh-1* mutants contained stacked oocytes and 82.3% showed no stacked quiescent oocytes (*n* = 96). In addition, the stacked oocytes of *gsa-1* RNAi-treated *fog-2; tbh-1* mutants are different from normal quiescent oocytes. They are bigger and rounder than those of *gsa-1* RNAi-treated *fog-2* mutants (Fig. 4b, *t* = 0). For two days, *gsa-1* RNAi-treated *fog-2* mutants maintained a similar number of quiescent oocytes, whereas the stacked oocytes of *gsa-1* RNAi-treated *fog-2; tbh-1* mutants were all exhausted (Fig. 4b, *t* = 21).

Figure 4a, b show the meiotic maturation and ovulation processes of a *gsa-1* RNAi-treated *fog-2* mutant and a *gsa-1* RNAi-treated *fog-2; tbh-1* mutant. As shown in Fig. 4b, at *t* = 0, this *gsa-1* RNAi-treated *fog-2; tbh-1* mutant contains 13 stacked oocytes. The first two oocytes (red arrows) are dark in color and round with no nuclear membrane, indicating they undergo maturation. Twelve hours later, 11 out of 13 stacked oocytes became mature or became Emo (orange arrow indicates the 11th mature oocyte). Twenty-one hours later, all 13 stacked oocytes became Emo. When we evaluated the maturation rate by counting Emo in a given duration as an indirect measurement of an ovulation rate, the rate of *gsa-1* RNAi-treated *fog-2; tbh-1* (0.53 ± 0.20, *n* = 14) is faster than that of *gsa-1* RNAi-treated *fog-2* (0.083 ± 0.043, *n* = 74), again showing *gsa-1* is not epistatic to *tbh-1*. Therefore, our data suggest that the OA signaling required for maintaining quiescent oocytes functions independently of the $G_{\alpha s}$ signaling that mediates oocyte maturation activated by sperm signal.

We observed that *fog-2; tbh-1* double mutants occasionally stacked oocytes in their gonad arms. To determine whether those stacked oocytes are quiescent and intact, we selected 1-d old females that contained stacked oocytes (42%, *n* = 31) and mated them with wild-type males. Mated *fog-2; tbh-1* mutants produced the Emo phenotype within 5 h of mating (76.5% Emo, *n* = 17), whereas mated *fog-2* females or OA-treated *fog-2; tbh-1* did not (0% Emo, *n* = 13) (Fig. 4c–e). 1-d old *fog-2* females rapidly ovulated stacked oocytes upon mating (17.7 ± 4.0 min/ovulation/ arm, *n* = 13) and no quiescent oocytes were left within 5 h of mating (Fig. 4c). In contrast, when 1-d old adult *fog-2; tbh-1* females with stacked oocytes were selected for mating, the stacked

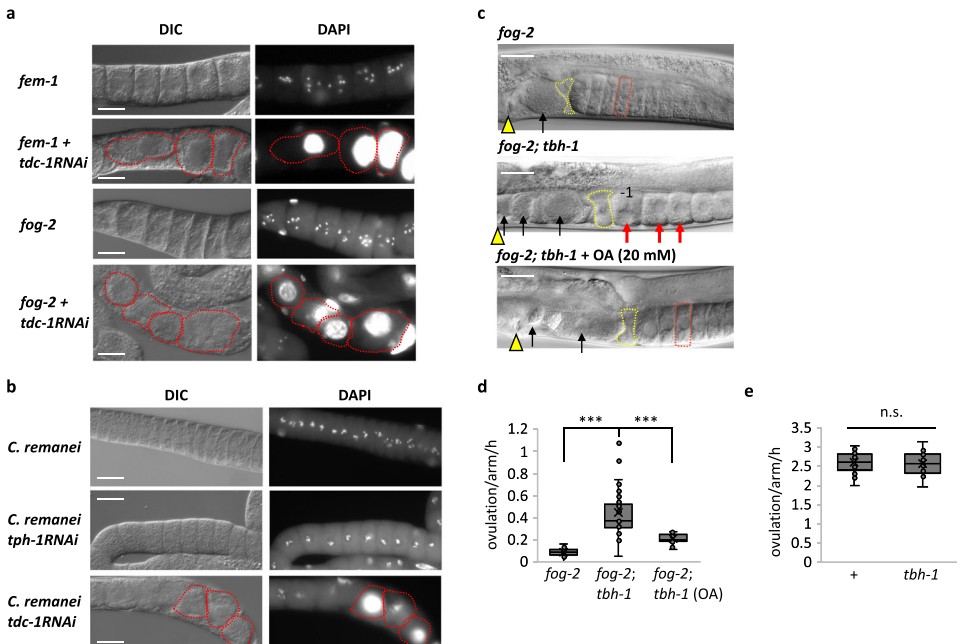

**Fig. 3 Maintenance of quiescent oocytes by octopamine in *Caenorhabditis* females. a** *fem-1* and *fog-2* mutants treated with control or *tdc-1* RNAi. Each image represents the 20 − 40 gonad images. Red-dotted circle indicates individual Emo. **b** A *C. remanei* female treated with control RNAi or *tdc-1* RNAi. *tph-1* encodes a tryptophan hydroxylase. Similar images were observed from 33 gonads for an empty vector, 36 gonads for *tph-1* RNAi and 23 gonads for *tdc-1* RNAi. **c** In *fog-2* mutants, 6 − 7 quiescent oocytes accumulate (red rectangle). Arrowheads: vulva, yellow polygons: spermatheca. In *fog-2; tbh-1* mutants, no quiescent oocytes accumulate. Due to a faster ovulation rate than that of *fog-2* mutants, three unfertilized oocytes in the uterus are shown (black arrows). The nucleus of the most proximal oocyte (−1) migrated to its cortex, indicating it is undergoing meiotic maturation[23]. Red arrows: activated oocytes. OA (20 mM) rescues the *fog-2; tbh-1* phenotype. Two unfertilized oocytes are shown in the uterus due to delay of rescue (black arrows). All animals were virgins observed 10 h after late L4. Similar images were observed from 7 gonads for *fog-2*, 16 gonads for *fog-2; tbh-1* and 9 gonads for *fog-2; tbh-1* + OA. **a–c** Scale bar = 20 μm. **d** Ovulation rates of *fog-2* (*n* = 28 animals), *fog-2; tbh-1* (*n* = 71 animals) and *fog-2; tbh-1* + OA (*n* = 35 animals). Two-sided *t*-test: *p* = $2.6 \times 10^{-15}$ (***) for *fog-2* vs. *fog-2; tbh-1*, $4.5 \times 10^{-10}$ (***) for *fog-2; tbh-1* vs. *fog-2; tbh-1* + OA. **e** Wild-type (+) (*n* = 20 animals) and *tbh-1* mutants (*n* = 14 animals) show similar ovulation rates in the presence of sperm. (n.s.: not significant, two-sided *t*-test: *p* = 0.76). **d**, **e** Boxplots show the median, mean (X), interquartile range (IQR). The upper whisker: the maxima smaller than 1.5 times IQR plus the third quartile, the lower whisker: the minima larger than 1.5 times IQR minus the first quartile. Source data are provided as a Source Data file.

oocytes became Emo within 5 h of mating (*n* = 12) (Fig. 4d, red arrows). The Emo phenotype of *fog-2; tbh-1* was rescued by OA (20 mM) restoring a similar rate of ovulation (18.9 ± 2.2 min/ovulation/arm, *n* = 11). Taking the delay in action of exogenously treated OA into account, we examined ovulation 14 h after the setup of mating.

This result suggests that the stacked oocytes in *fog-2; tbh-1* were not quiescent and that the sperm signal that promotes ovulation rate in *fog-2; tbh-1* females allowed us to visualize the defect. Approximately 40% embryos of the mated *fog-2; tbh-1* females were unable to hatch (Fig. 4g), suggesting the ill-timed awakening of oocytes in the absence of OA contributes to abnormal fertilization and/or development. When we mated *fog-2; tbh-1* females at late L4 stage, however, they reproduced normally; they did not produce Emo and the most embryos hatched (Fig. 4f, g). This indicates the oocytes of *fog-2; tbh-1* are fully functional as long as they do not need to enter quiescence. When we mated the *tdc-1* RNAi-treated *C. remanei* females at late L4 stage, they did not produce Emo, either. These results indicate that once oocytes fail to maintain quiescence, introducing sperm afterwards lowers the chance to produce viable progeny. These results support that for these *Caenorhabditis* species, OA functions as a safeguard to maintain quiescent oocytes and is critical before the sperm signal becomes available.

### Maintenance of quiescent oocytes by OA in *D. melanogaster*.
To test whether OA serves as a conserved signal for oocyte quiescence, we examined *D. melanogaster*. In *D. melanogaster*, previtellogenic egg chambers consist of an oocyte and 15 nurse cells, surrounded by a layer of somatic follicle cells[44]. During winter, fruit flies in the wild enter reproductive diapause or dormancy and do not contain egg chambers past stage 8[14,15,21,22]. This diapause is considered different from reduced oogenesis by starvation, during which the proliferation rate of follicle cells reduces to ¼ of that of well-fed condition[14,45,46]. Starved previtellogenic egg chambers share certain defined characteristics of oocyte quiescence in other animal germlines such as redistribution of ribonucleoprotein complex components and cortically condensed microtubules[46,47]. Based on this similarity, hereafter we call reduction of oogenesis or temporary developmental arrest of previtellogenic egg chambers induced by starvation as oocyte quiescence.

We examined OA function in maintaining oocyte quiescence under protein starvation by removing yeast components from the media. Without yeast, the major source of protein[48], the synthesis of yolk proteins is reduced and the egg chambers grow very slowly[45]. This quiescence of egg chambers is rapidly reversed by a protein-containing diet (Fig. 5a)[47,49]. We examined the ovaries of virgins from immediately after they eclosed, in wild-type (Canton-S or $w^{1118}$), $tbh^{nM18}$ (hereafter called $tbh^{-/-}$) (a null allele that does not produce OA)[50], and $tbh^{+/nM18}$ flies (hereafter called $tbh^{+/-}$) at 0-day post-eclosion (*dpe*), 1 *dpe*, 1.5 *dpe*, and 2 *dpe* upon protein starvation (1 *dpe* is 24 h-, 1.5 *dpe* is 36 h-, 2 *dpe* is 48 h-post-eclosion). We picked those time points because flies do not lay eggs until 2 *dpe*. This is necessary to examine the sole

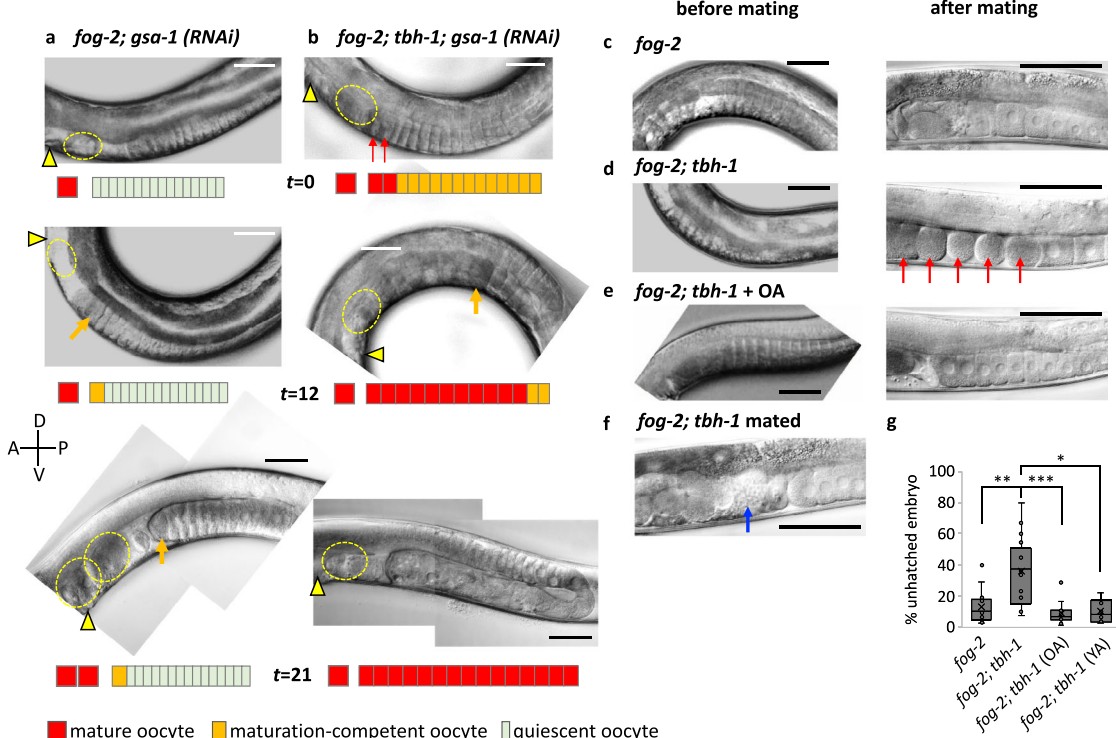

**Fig. 4 Octopamine acts in parallel to sperm signaling. a, b** OA negatively regulates ovulation in the absence of sperm signal independent of $G_{\alpha s}$ signaling. The ovulation process of a virgin of *fog-2* with *gsa-1* RNAi (**a**) or *fog-2; tbh-1* with *gsa-1* RNAi (**b**) was observed at 36 h ($t = 0$), 48 h ($t = 12$), and 57 h ($t = 21$) after the late L4 stage. Triangle indicates vulva, red arrows mature oocytes, and yellow dotted circle a matured oocyte ovulated in the uterus or trapped inside the spermatheca. At time point 0, oocytes were stacked in both *fog-2; gsa-1RNAi* and *fog-2; tbh-1; gsa-1RNAi* ($n = 12$ gonads for both). **c–e** Representative images of *fog-2* and *fog-2; tbh-1* before and after mating (5 h) and OA treated. Similar images were observed from 13 animals for *fog-2*, 12 animals for *fog-2; tbh-1* and 11 animals for *fog-2; tbh-1* + OA. Red arrow: Emo. **f** Mating does not produce Emo when late L4 females of *fog-2; tbh-1* mutants (similar images were observed from 6 animals) were mated with wild-type males. We observed 98 ovulations occurring in 25 h (~30 min/ovulation/arm). Sperm are indicated by the blue arrow. **a–f** Scale bar = 50 μm. **g** Percent unhatched embryos. *fog-2* ($n = 400$ embryos), *fog-2; tbh-1* ($n = 358$ embryos), *fog-2; tbh-1* (OA) ($n = 948$ embryos) and *fog-2; tbh-1* (YA) ($n = 437$ embryos). Two-sided *t*-test: $p = 0.0012$ (**) for *fog-2* vs. *fog-2; tbh-1*, $p = 0.0001$ (***) for *fog-2; tbh-1* vs. *fog-2; tbh-1* + OA, $p = 0.28$ for *fog-2* vs. *fog-2; tbh-1* + OA. Boxplots show the median, mean (X), interquartile range (IQR). The upper whisker: the maxima smaller than 1.5 times IQR plus the third quartile, the lower whisker: the minima larger than 1.5 times IQR minus the first quartile. Source data are provided as a Source Data file.

effect of OA on oogenesis because *tbh⁻/⁻* mutants are defective in egg-laying[24] and accumulation of full-grown stage 14 egg chambers would interfere with the analysis of the phenotypes if we examine the oogenesis after 2 *dpe*. At 0 *dpe*, immediately after eclosion, most of the ovaries of all flies contained stage 6/7 egg chambers. The size and the number of total egg chambers of *tbh⁻/⁻* flies were similar to those of *w¹¹¹⁸* flies, although due to a background difference, Canton-S contains fewer egg chambers (Fig. 5b). At 1 *dpe*, however, most ovaries of *tbh⁻/⁻* contained further grown egg chambers, such as stage 10B or 12 egg chambers, whereas most *tbh⁺/⁻* ovaries contained mainly small-sized stage 8 egg chambers (Fig. 5c, d). At 1.5 *dpe* and 2 *dpe*, most *tbh⁻/⁻* ovaries contained many full-grown stage 14 egg chambers, whereas a few *tbh⁺/⁻* ovaries contained stage 14 egg chambers (Fig. 5c, e, f). This result suggests that egg chambers stop growing upon starvation in an OA-dependent manner and that accumulation of stage 14 egg chambers in *tbh⁻/⁻* mutants is due to failure to maintain quiescent egg chambers.

In *D. melanogaster*, there are four G-protein-coupled OA receptors. OAMB (OctopAmine receptor in Mushroom Bodies) is similar to vertebrate α-adrenergic receptors and essential for egg-laying[30], whereas the other three (OA2/Octβ1R, Octβ2R, Octβ3R) are similar to β-adrenergic receptors[51]. Among the three β-adrenergic receptors, we tested a mutant of *octβ2R*, *octβ2R^f05679*. An *in situ* hybridization study showed *octβ2R* expression in the

nurse cells of previtellogenic egg chambers[52], suggesting it might be a receptor for the OA action in oocyte quiescence. The allele *octβ2R^f05679* contains a piggyBac insertion and is considered to be a significantly reduced-function mutant[53]. The mutant shows a similar phenotype to *tbh⁻/⁻*; at 2 *dpe* upon starvation, their ovaries contained mainly stage 14 egg chambers (Fig. 6a, b). Exogenous OA (5 mg/ml) rescued the unrestrained growth of egg chambers in the *tbh⁻/⁻* mutant upon starvation, confirming that lack of OA caused the defect (Fig. 6c)[50]. Exogenous OA could not rescue the defect of *octβ2R^f05679* (*octβ2R⁻/⁻*) mutants, however, supporting the idea that Octβ2R is a receptor for OA because without the receptor, adding the ligand will not rescue the phenotype. Together, our data suggest OA is a signal to maintain oocyte quiescence during protein starvation in *D. melanogaster*.

Next, we asked how OA signaling balances with nutrient signaling and whether OA can inhibit oocyte growth competing with nutrient signals when, for instance, the nutrients are insufficient to support rapid oogenesis. To address this, we depleted nutrients completely and then counted the number of stage 14 egg chambers at 2 *dpe* varying nutrient levels. Under the no nutrient condition, control virgins produce no or few stage 14 egg chambers, whereas *tbh⁻/⁻* virgins produce stage 14 egg chambers at a similar rate to that of Canton-S virgins fed on CSY standard media (Fig. 6d. Compare *tbh⁻/⁻* in Fig. 6d to 2.5% in Fig. 6f). *tbh⁺/⁻* virgin flies produce numbers of stage 14 egg

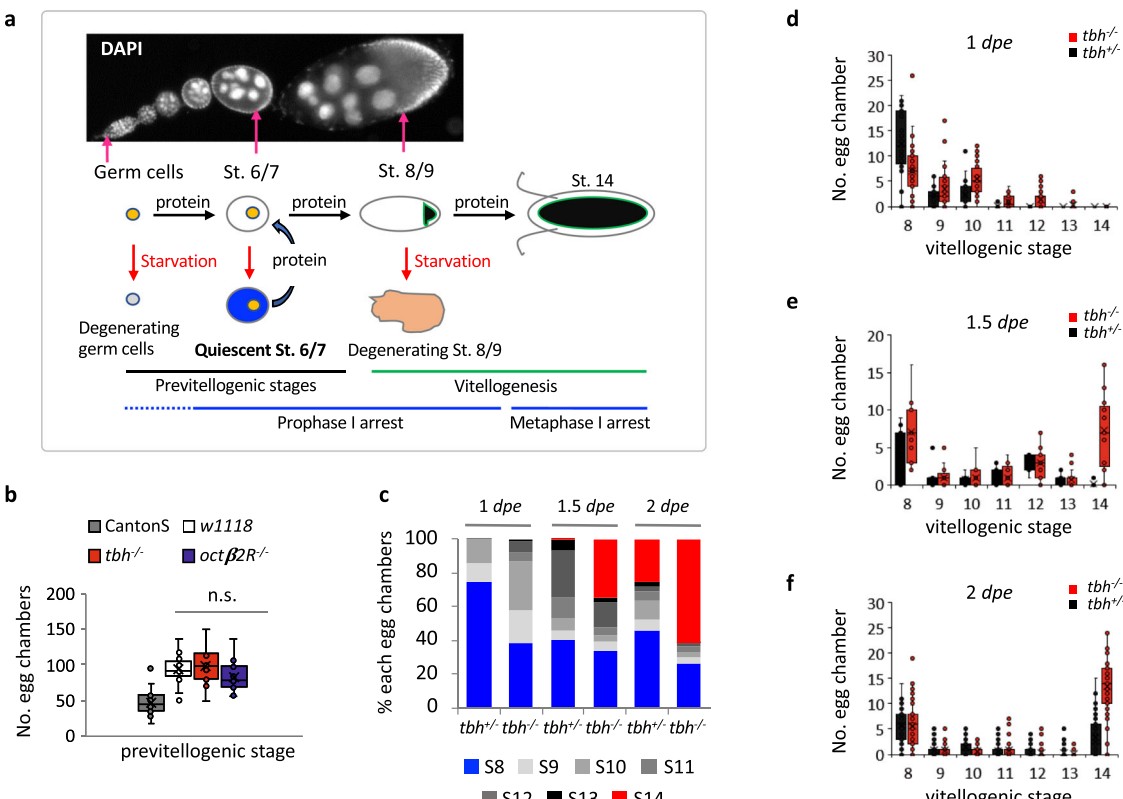

**Fig. 5 Octopamine is required for maintaining previtellogenic egg chambers upon nutrient deprivation in *D. melanogaster*. a** Oogenesis of *D. melanogaster*. Upon protein starvation, either germ cells or stage 8/9 egg chambers are degenerated. The previtellogenic egg chambers slow down the growth, then resume the growth within 2 h once protein is available. During egg chamber growth, meiotic chromosomes arrest at the prophase I until stage 13 (blue lines). **b** The number of previtellogenic egg chambers (stages 2-7) per fly are similar between wild-type (*w¹¹¹⁸*) female virgin controls ($n = 13$ animals) and the octopamine mutant female virgins (*tbh⁻/⁻* ($n = 11$ animals) and *octβ2R⁻/⁻* ($n = 14$ animals)). Two-sided *t*-test: $p = 0.63$ (*w¹¹¹⁸* vs. *tbh⁻/⁻*), 0.22 (*w¹¹¹⁸* vs. *octβ2R⁻/⁻*), 0.12 (*tbh⁻/⁻* vs. *octβ2R⁻/⁻*). Canton-S ($n = 35$) has fewer number of egg chambers than *w¹¹¹⁸*. $p = 5.5 \times 10^{-11}$ (Canton-S vs. *w¹¹¹⁸*), $1.7 \times 10^{-10}$ (Canton-S vs. *tbh⁻/⁻*), $1.8 \times 10^{-8}$ (Canton-S vs. *octβ2R⁻/⁻*). n.s.: not significant. **c** The percent of each stage of egg chambers in the ovaries of *tbh⁺/⁻* and *tbh⁻/⁻* virgins during the protein starvation period. The percent of stage 14 egg chambers increases in *tbh⁻/⁻* mutants compared to that of the heterozygote control, *tbh⁺/⁻*. These data are extracted from the raw data shown in **d–f**. **d–f** The distribution of each stage egg chambers in the ovaries of each *tbh⁺/⁻* ($n = 29$ animals for 1 dpe, 14 animals for 1.5 dpe, 59 animals for 2 dpe) and *tbh⁻/⁻* ($n = 32$ animals for 1 dpe; 17 animals for 1.5 dpe, 59 animals for 2 dpe) female virgin during the protein starvation period. At each time point, *tbh⁻/⁻* mutants contain more progressed egg chambers compared to *tbh⁺/⁻*. **b, d–f** Boxplots show the median, mean (X), interquartile range (IQR). The upper whisker: the maxima smaller than 1.5 times IQR plus the third quartile, the lower whisker: the minima larger than 1.5 times IQR minus the first quartile. Source data are provided as a Source Data file.

chambers that range between wild-type and *tbh⁻/⁻*, suggesting haploinsufficiency of OA (Fig. 6d). Exogenous OA restores oocyte arrest to *tbh⁺/⁻* and *tbh⁻/⁻* mutants in a concentration dependent manner (Fig. 6e). When we varied the level of nutrients using 2.5% yeast (CSY2.5, rich media) or 1% yeast (CYS1%, poor media), flies on CSY2.5% produced more stage 14 egg chambers than those on CSY1% (Fig. 6f), indicating that the number of stage 14 egg chambers increases as nutrients increase. Exogenous OA (5 mg/ml) inhibits the production of stage 14 egg chambers on poor media but not on rich media, suggesting rich nutrients override the inhibitory signal of OA (Fig. 6f). However, even on rich media, the production of stage 14 egg chambers is reduced at a higher concentration of OA (10 mg/ml) (Fig. 6f: 2.5% + OA10). These results demonstrate the role of OA in balancing nutrient signaling to maintain quiescent egg chambers.

**Maintenance of quiescent oocytes by NE in *D. rerio*.** Next, we asked whether noradrenergic signal in a vertebrate also plays a role in maintaining quiescent oocytes by examining zebrafish ovaries after prolonged starvation. The five stages of oocytes are

easily distinguishable based on size and shape[12]; the stage I oocytes are small, transparent (<0.14 mm in diameter), and encapsulated by a layer of pre-granulosa cells forming primordial follicles. At the beginning of stage II, protein- and carbohydrate-containing small cortical alveoli start to accumulate, which occupy much of the ooplasm as the oocyte reaches the later stage II with a diameter of 0.27–0.34 mm (Fig. 7a). Stage III is the major growing stage where the oocytes become opaque due to vitellogenesis and the diameter of an oocyte reaches up to 0.69 mm. The oocytes mature through stage IV and become eggs (stage V) with a diameter around 0.75 mm[12].

Zebrafish contains a single *dbh* gene to convert dopamine to NE (Fig. 1a). To investigate the NE function in oocyte quiescence, we examined a *dbh⁻/⁻* allele that contains a four nucleotide insertion, which causes a premature stop codon and produces non-functional enzymes[26]. When we compared oogenesis between *dbh⁻/⁻* and *dbh⁺/⁻* females after 24 days of starvation, *dbh⁺/⁻* ovaries contained mostly primordial follicles of stage I, a small fraction (about 5% of the total follicles) of activated previtellogenic follicles of late stage II and several mature oocytes (stage IV). However, most ovaries did not contain vitellogenic

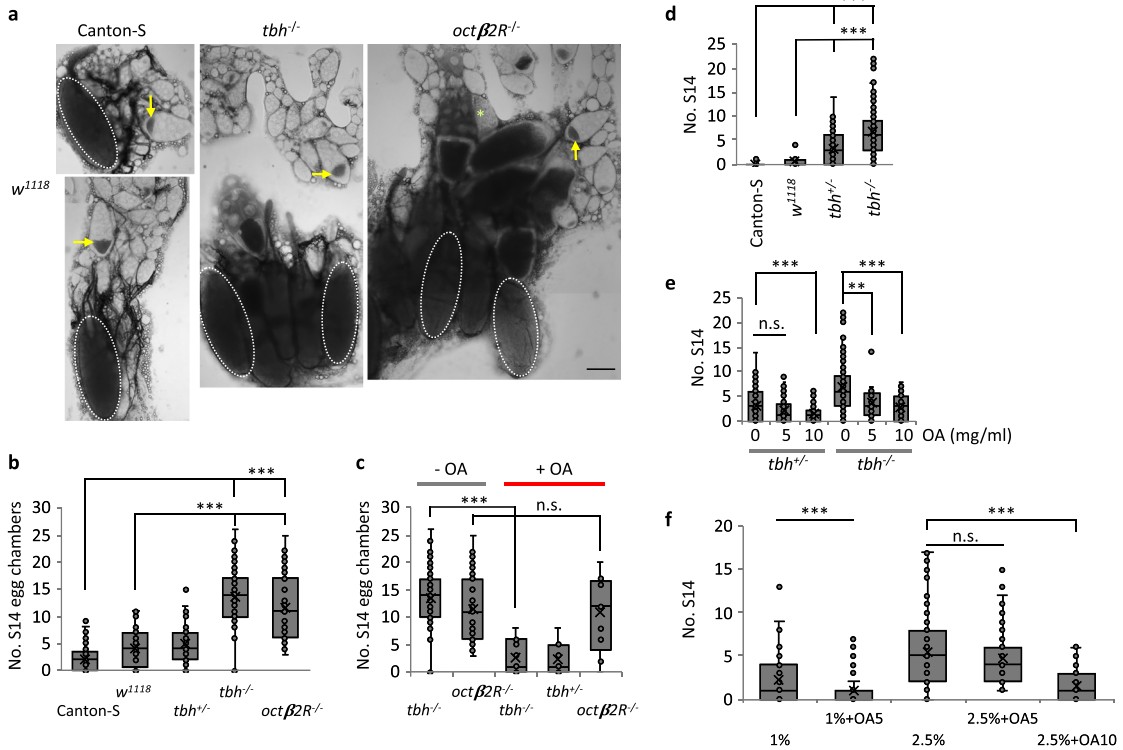

**Fig. 6 Octopamine regulates quiescence via Octβ2R, competing with nutrients. a, b** Upon protein starvation, $tbh^{-/-}$ or $oct\beta2R^{-/-}$ virgins contain more stage 14 egg chambers (S14, white dotted ellipses) than controls. Representative images (Scale bar = 100 μm) (**a**) and the quantification (**b**). Arrows indicate stage 9. $n = 85$ $tbh^{-/-}$; 35 $oct\beta2R^{-/-}$; 137 Canton-S; 52 $w^{1118}$; 47 $tbh^{+/-}$. Two-sided $t$-test: $p =$ CS vs. $w^{1118}$ (0.0001); vs. $tbh^{+/-}$ (1.2 × 10$^{-6}$); vs. $tbh^{-/-}$ (8.4 × 10$^{-36}$); vs. $oct\beta2R^{-/-}$ (1.4 × 10$^{-10}$), $w^{1118}$ vs. $tbh^{+/-}$ (0.21); vs. $tbh^{-/-}$ (2.6 × 10$^{-25}$); vs. $oct\beta2R^{-/-}$ (2.3 × 10$^{-10}$); 7.2 × 10$^{-21}$ ($tbh^{-/-}$ vs. $tbh^{+/-}$); 8.8 × 10$^{-7}$ ($tbh^{+/-}$ vs. $oct\beta2R^{-/-}$); 0.070 ($tbh^{-/-}$ vs. $oct\beta2R^{-/-}$). **c** Exogenous OA (5 mg/ml) rescues $tbh^{-/-}$ mutants, but not $oct\beta2R^{-/-}$. $n = 85$ animals (−OA), 11 animals (+OA) for $tbh^{-/-}$; 35 animals (−OA), 9 animals (+OA) for $oct\beta2R^{-/-}$; 7 animals (+OA) for $tbh^{+/-}$. Two-sided $t$-test: $p = 7.5 × 10^{-9}$ for $tbh^{-/-}$, 0.77 for $oct\beta2R^{-/-}$ between untreated and treated with OA. The same data for without OA in Fig. 6b was replotted. **d** Without nutrients, controls produce few S14 ($n = 25$ Canton-S, 11 $w^{1118}$), whereas $tbh^{-/-}$ ($n = 99$ animals) produces at a similar rate to that of Canton-S fed on CSY. $tbh^{+/-}$ ($n = 116$ animals) produces S14 ranging between wild-type and $tbh^{-/-}$. Two-sided $t$-test: $p =$ CS vs. $w^{1118}$ (0.05), vs. $tbh^{+/-}$ (3.1 × 10$^{-6}$), vs. $tbh^{-/-}$ (1.7 × 10$^{-10}$), $w^{1118}$ vs. $tbh^{+/-}$ (0.008), vs. $tbh^{-/-}$ (4.4 × 10$^{-5}$), 9.4 × 10$^{-10}$ ($tbh^{+/-}$ vs. $tbh^{-/-}$). **e** Exogenous OA restores oocyte quiescence to $tbh^{+/-}$ and $tbh^{-/-}$ in a concentration dependent manner. $n = 33$ $tbh^{+/-}$, 28 $tbh^{-/-}$ for 5 mg/ml OA, 43 $tbh^{+/-}$, 26 $tbh^{-/-}$ for 10 mg/ml OA. Two-sided $t$-test: $p = 9.4 × 10^{-10}$ ($tbh^{+/-}$ vs. $tbh^{-/-}$), 0.11 ($tbh^{+/-}$: OA0 vs. OA5), 0.0005 ($tbh^{-/-}$: OA0 vs. OA5), 0.0013 ($tbh^{+/-}$: OA0 vs. OA10), 0.0001 ($tbh^{-/-}$: OA0 vs. OA10). **f** OA and nutrients compete in regulation of egg chamber growth. $n = 55$ Canton-S (CSY1%), 73 Canton-S (1% + OA5), 107 Canton-S (CSY2.5%), 79 Canton-S (CSY2.5% + OA5), 53 Canton-S (CSY2.5% + OA10). Two-sided $t$-test: $p = 0.0006$ (CSY1% vs. CSY1% + OA5), 0.21 (CSY2.5% vs. CSY2.5% + OA5), 1.4 × 10$^{-7}$ (CSY2.5% vs. CSY2.5% + OA10). **b–f** Numbers of egg chambers per fly were shown. Boxplots show the median, mean ($X$), interquartile range (IQR). The upper whisker: the maxima smaller than 1.5 times IQR plus the third quartile, the lower whisker: the minima larger than 1.5 times IQR minus the first quartile. Source data are provided as a Source Data file.

follicles of stage III (Fig. 7b, d, f, g). This observation suggests that under long-term starvation, most primordial follicles become quiescent in $dbh^{+/-}$ ovaries and only a small fraction of total follicles undergo PFA. After 45-day starvation, the $dbh^{+/-}$ ovaries contain only stage I follicles (Fig. 7h), suggesting that follicles in other stages eventually disappear and only the stage I oocytes remain quiescent as starvation continues.

In contrast, after 24 days of starvation, $dbh^{-/-}$ ovaries contain many late stage II and vitellogenic stage III follicles (Fig. 7c, red-dotted circle as examples). $dbh^{-/-}$ ovaries contain stage II follicles even after 32 days of starvation (about 18% of the total follicles, Fig. 7e–g). Most strikingly, every starved $dbh^{-/-}$ ovary we examined contained early stage III follicles ($n = 22$), which we rarely observed in $dbh^{+/-}$ ($n = 14$). This suggests that in the absence of NE the follicles keep growing and reach the early vitellogenic stage, which does not happen in the presence of NE. Taken together, these results suggest that NE is required for maintaining oocyte quiescence in zebrafish.

After three-weeks of starvation, late stage III oocytes (0.34 − 0.69 nm in diameter) are absent in both $dbh^{-/-}$ and

$dbh^{+/-}$ ovaries (Fig. 7b–e). This suggests that as in flies, if the starvation happens after the follicle has passed stage III, the follicle degenerates due to lack of building blocks and energy. The fact that late stage III oocytes are absent in both $dbh^{-/-}$ and $dbh^{+/-}$ ovaries also indicates that NE is not required for degeneration of oocytes.

Consistent with a previous report, $dbh^{-/-}$ female zebrafish reproduce normally under well-fed conditions[26]. $dbh^{-/-}$ mutant females produce 100 − 300 eggs upon mating, and the ovaries do not show any morphological differences from those of wild-type. Also, starvation-induced quiescence of oocytes is reversible; upon refeeding, the starved fish resumed oogenesis within 2 days.

## Discussion

Maintaining quiescent oocytes is pivotal for quality reproduction span. We identified the noradrenergic signal as a conserved oocyte quiescence signal in two *Caenorhabditis* species while the oocytes are waiting for sperm, and in *D. melanogaster* and *D. rerio* when they are deprived of nutrients. The ovaries of all four

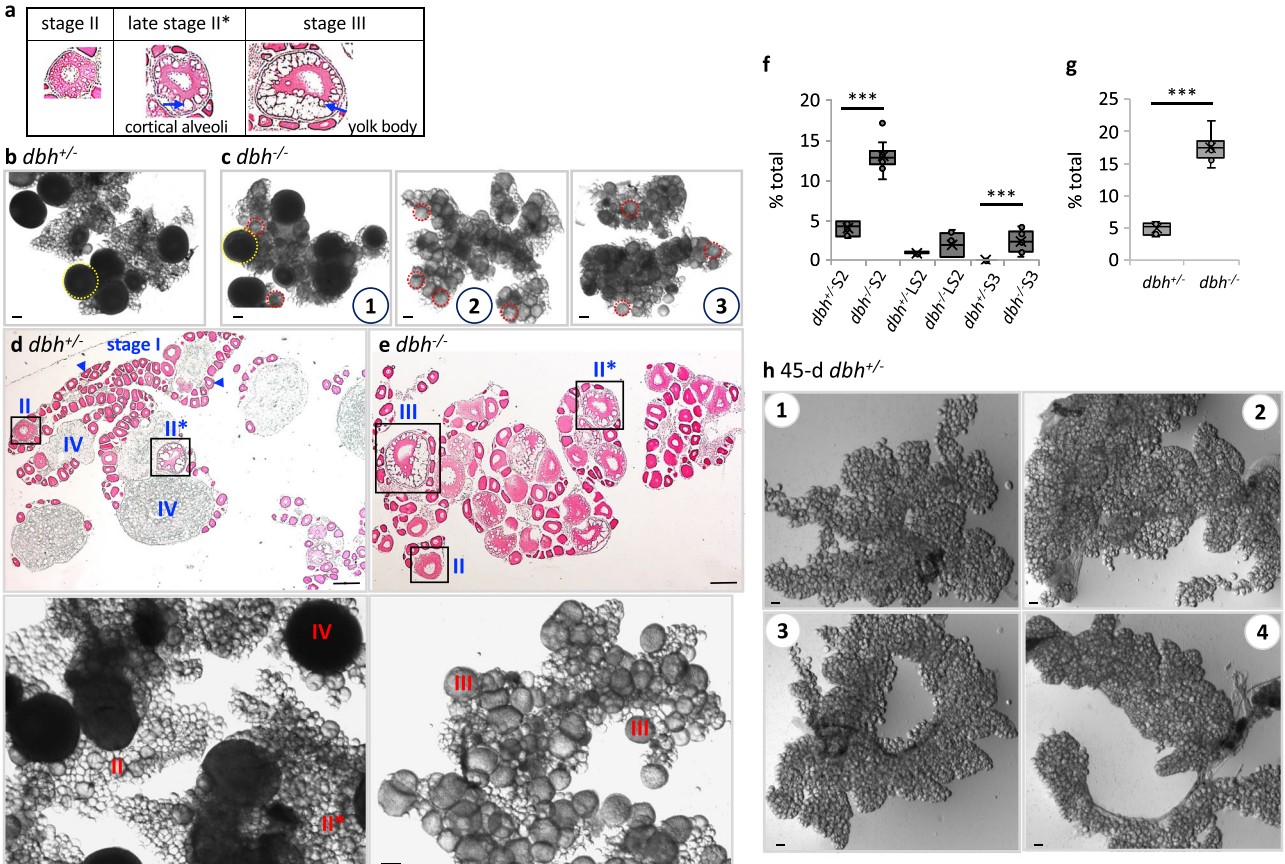

**Fig. 7 NE is necessary for maintenance of quiescent oocytes during starvation. a** Follicle images of each stage of oocytes based on size and shape. **b** A representative image of an ovary of 24-d starved $dbh^{+/-}$ shows two major populations of oocytes of stage IV (dark and big, yellow circle) and stage I (transparent and small). Similar images were observed from 9 samples. **c** Images of three different ovaries of 24-d starved $dbh^{-/-}$. In contrast to $dbh^{+/-}$, most oocytes are stage II or III (intermediate size, red circle). Similar images were observed from 14 samples. **d** Upper panel: a representative image of an ovary of 24-d starved $dbh^{+/-}$ stained with neutral red. Stage I oocytes are most abundant (arrowheads). Oocytes in each stage of II, II* (late stage II), and IV are easily visible. Lower panel: an image of the whole ovary. **e** A representative image of an ovary of 32-d starved $dbh^{-/-}$. Upper panel: neutral-red stained image. Compared to $dbh^{+/-}$, oocytes of later stages (e.g., stages II and III) are abundant. Lower panel: an image of the whole ovary. **b–e** Scale bar = 200 μm. **f** The percent of oocytes at stage II (S2), late stage II (LS2), and stage III (S3) in $dbh^{+/-}$ ($n = 8$ sections from three independent fish) and $dbh^{-/-}$ ($n = 10$ sections from four independent fish). The stage III oocytes were only observed in $dbh^{-/-}$. Two-sided t-test: $p = 3.1 \times 10^{-9}$ between $dbh^{+/-}$ vs. $dbh^{-/-}$ in stage II; $p = 0.04$ in late stage II; $p = 0.0004$ for stage III. **g** The sum of stage II, late stage II, and stage III oocytes presented in **f**. Two-sided t-test: $p = 2.3 \times 10^{-11}$. **f–g** Boxplots show the median, mean ($X$), interquartile range (IQR). The upper whisker: the maxima smaller than 1.5 times IQR plus the third quartile, the lower whisker: the minima larger than 1.5 times IQR minus the first quartile. **h** Ovaries of four different $dbh^{+/-}$ females after 45 days of starvation contain only stage I oocytes. Ovaries of four other females showed similar phenotype. Scale bar = 100 μm. Source data are provided as a Source Data file.

species are heavily innervated by cells producing OA or NE. Failure of oocyte quiescence results in endomitotic oocytes, and then sterility in the two *Caenorhabditis* species, unrestrained growth of egg chambers in *D. melanogaster*, and excess activation of primordial follicles in *D. rerio*.

In *C. elegans*, OA is essential for maintaining quiescent oocytes when the sperm signal is absent. We observed that in *fog-2; tbh-1* mutants, OA is not required if the sperm signal is present early enough to prevent accumulation of quiescent oocytes; both the *tbh-1* mutant hermaphrodite and the *fog-2; tbh-1* females mated from L4s reproduce normally without producing Emo phenotypes. *C. elegans* oocytes are not individually surrounded by somatic follicle cells as in mammals. Instead, they grow inside the gonadal sheath communicating with them via gap junctions, which therefore serves a function similar to the follicle cells in mammals for maintaining quiescent oocytes. In *ceh-18* or *inx-14/22* mutants, which show abnormal sheath differentiation or lack of gap junctions to oocytes, respectively, quiescent oocytes are not maintained, resulting in faster ovulation and frequent Emo in the

absence of sperm[37,39,54]. In addition, certain G-protein signals, such as $G_{\alpha o}$, are known to play a role in the gonadal sheath in maintaining oocyte quiescence[54]. As our results suggest that OA signal does not directly interact with $G_{\alpha s}$, which mediates the sperm signal, we suggest that OA from gonadal sheath cells may act on SER-3, a G-protein-coupled receptor, in an autocrine fashion and produce downstream inhibitory signals which are transported to the oocytes through gap junctions and maintain oocyte quiescence.

Upon nutrient deprivation, *D. melanogaster* virgins slow down the growth of previtellogenic egg chambers, which remain quiescent until nutrients are available. Our data suggest that OA is required for maintaining oocyte quiescence. $tbh^{-/-}$ mutants could sense D-glucose in gustatory sensory neurons upon starvation and consume the same amount of food as wild-type under both fed and starved conditions, showing that the hunger sensation in the mutant is intact[55,56]. This excludes the possibility that OA may be directly involved in sensing nutrient homeostasis and the mutants fail to evaluate the nutrient condition and, in

turn, fail to maintain oocyte quiescence. Rather, OA appears to regulate oocyte quiescence directly, as the ovaries have access to OA from OA-expressing neurons innervating the ovarian epithelium[51,57,58].

Nutrient-sensing pathways play a major role in promoting the growth of egg chambers and follicles; in flies, a high level of dILPs and JH promote the growth of the egg chamber[19,22]. In mammals, the mTOR and the insulin-FOXO pathways coordinate to awaken quiescent oocytes[4]. The fact that nutrient-sensing pathways regulate the timing of oocyte awakening suggests that deprivation of nutrients would do the opposite and stop oogenesis. Indeed, a reduced level of insulin signal halts the growth of egg chambers in flies[22]. It remains unclear, however, whether the absence of a nutrient signal is sufficient to maintain quiescent oocytes. Our results that the nutrient signal and OA signal could balance each other suggest that when the level of nutrients is insufficient, OA signal could be critical to maintain quiescent egg chambers.

Although our zebrafish study was based on observations made from a mutant and we have not identified the receptor(s) and the action site in the ovary, we could observe that the quiescent oocytes were maintained in the control whereas they were not in the mutant fish, which is the same as we observed in the invertebrate animals that we examined. Based on these results, we suggest that the nutrient-responsive neuronal signals are necessary for promoting and stopping the oogenesis and that OA/NE are necessary for maintaining the quiescent oocytes once the nutrient-responsive signals are sufficiently reduced. The ovary receives noradrenergic input in both vertebrates and invertebrates; NE is released from the sympathetic nerves innervating ovary and regulates steroidogenesis and early follicular development in mammals[59,60]. In addition, starvation activates sympathetic nerves[61] and thus would lead to release of NE into the ovary. OA production in *C. elegans* is also increased under starvation[62]. These observations may suggest that unfavorable conditions for reproduction such as starvation would increase noradrenergic input in the ovary.

Based on our results, we propose a model of noradrenergic signaling in oocyte quiescence (Fig. 8). In *C. elegans*, quiescent oocytes are maintained by OA. Sperm override OA's maintenance signal and induce oocyte maturation. Nutrients probably would not play a significant role in controlling oocyte maturation because, unlike other animals, reduced vitellogenesis in *C. elegans* does not severely hamper embryogenesis[63]. In *D. melanogaster* and *D. rerio*, quiescent oocytes are maintained by OA/NE in the absence (or at low level) of nutrient signals. Availability of nutrients overrides this maintenance signal, and oocytes grow and mature (Fig. 8a). Our results in *Drosophila* and zebrafish suggest that the onset of oogenesis should be regulated by a delicate balance between two signaling systems: signals for abundant nutrients mediated by insulin and amino acids and signals for nutrition deprivation by the noradrenergic signal. The role of OA/NE would be the most relevant and critical when nutrients are insufficient for continuous oogenesis (Fig. 8b). OA/NE serves as a counterbalance signal to nutrients; increased levels of OA/NE causes oogenesis to stop, and oocytes become quiescent waiting for conditions to improve. In the wild, where availability of food is unpredictable, maintaining oocyte quiescence or releasing them from quiescence has to be determined after careful integration of the two signals. It is intriguing to speculate that OA/NE as an inhibitory signal safeguards and reins in oocyte growth so that even if the condition becomes temporarily better, the oocytes do not instantly commit to full throttled growth unless good conditions persist. If our model is correct then, when the ratio of OA/NE signal to nutrient signals is abnormally low, the imbalance would activate primordial follicles to grow even under unfavorable conditions and lead to exhaustion of the oocyte pool in mammals. Interestingly, it is reported that many polycystic ovarian syndrome (PCOS) patients who show an increase in the number of antral follicles, abnormal ovulation, and occasional sterility have either an excessive amount of NE with lower β-adrenergic receptor activity[64] or an undetectable amount of NE[59,64,65]. Considering many PCOS patients also suffer from diabetes or obesity-related syndromes, it is possible that they have an abnormally low ratio of NE to nutrient signals, which might contribute to development of PCOS.

Due to lack of directly connected blood vessels or sympathetic nerves, primordial follicles in vertebrates receive nutrients and NE by diffusion from stromal blood vessels and nearby neuronal varicosities, respectively[66]. Mice lacking NE are fertile and do not show severe reproductive problems[27]. Our study suggests that the role of NE is hidden under well-fed conditions, where a high level of nutrient signal would override NE's inhibitory signal. Interestingly, calorie-restricted female mice and rats show a significantly lower proportion of the early stage of the growing follicles and a higher proportion of the quiescent primordial follicles than well-fed animals[67,68]. As antral follicles are the major stage that degenerates during starvation, the higher

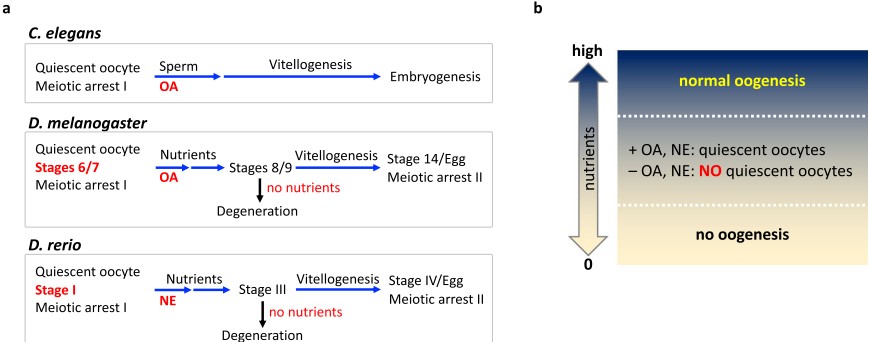

**Fig. 8 The model. a** The summary of oogenesis in different models shows the step and the stage where the noradrenergic input is required to maintain oocyte quiescence. Oocyte quiescence is maintained by OA. Quiescent oocytes are awakened by sperm in *C. elegans* and by nutrients in *D. melanogaster*. In zebrafish, NE would be overridden by nutrients, which serve as a signal of PFA. Under starvation conditions, early stage vitellogenic oocytes are subjected to degeneration in *D. melanogaster* and zebrafish. **b** For flies and zebrafish, oogenesis is tightly regulated by availability of food. When nutrients are completely deprived (0), oogenesis does not initiate due to lack of building blocks and energy. When nutrients are abundant (high), oogenesis progresses in full speed overriding any inhibitory signals such as NE/OA. When the nutrient level is insufficient for continuous oogenesis, NE/OA serves as a counterbalance signal to nutrients; oogenesis stops, and oocytes become quiescent waiting for the condition to be improved.

proportion of quiescent primordial follicles in calorie-restricted animals suggests reduction of awakened primordial follicles due to a low transition rate from primordial to growing follicles. Thus, it will be intriguing to examine oogenesis of mice lacking NE under these starvation conditions.

## Methods

**Animal strains and culture conditions.** *Caenorhabditis elegans* were routinely grown on nematode growth media (NGM) seeded with *E. coli* strain HB101 at 20 °C. The wild-type *C. elegans* was Bristol N2. Mutant strains used were BS553 *fog-2(oz40) V*, CB4108 *fog-2(q71) V*, DA1774 *ser-3(ad1774) I*, DA2142 *fem-1(hc17ts) IV*, MT9455 *tbh-1(n3247) X*, MT9971 *nIs107 [Ptbh-1::GFP + lin-15(+)] III*, MT13113 *tdc-1(n3419) II*. YJ249 *fog-2 (q71) V; tbh-1 (n3247) X* was generated as follows: *fog-2(q71)* males were crossed to *tbh-1(n3247)* hermaphrodites. The F$_1$ males were crossed to *tbh-1(n3247)* hermaphrodites, and ~100 L4 of F$_2$ hermaphrodites were plated individually. Females of *fog-2(q71) V; tbh-1(n3247) X*, were identified by *fog-2* phenotype and crossed with *fog-2* males to generate males of *fog-2(q71) V; tbh-1(n3247) X*. NGM plates containing 20 mM octopamine (OA) were used during crosses.

*Drosophila* strains, Canton-S, *w^1118^*, *tbh^nM18^*/FM7 (a gift from Dr. Monastirioti), *tdc2^RO54^* (a gift from Dr. Hirsh), *Tdc2-GAL4* (#9313), *UAS-GFP* (#4776), *UAS-mCD8::GFP*, and *octβ2R^f05679^* (#18896) (from Bloomington Drosophila stock center), were used. The strains were grown on the standard cornmeal-sugar-yeast media (CSY), containing 5.2 g cornmeal, 11 g table sugar, 2.5 g yeast (MP Biomedicals, Solon, OH), 1.1 ml of 20% tegosept, 0.5% propionic acid and 0.79 g agar per 100 ml of H$_2$O. The stocks were maintained by setting up matings between 30 females and 30 males.

*Danio rerio* strains used were *dbh^ct806^/+* (AB/TL) and Tg(*dbh:EGFP*)(gifts from Dr. Prober)[26]. Zebrafish were maintained in environmentally controlled rooms at the Bioscience and Biotechnology Center, Nagoya University, on 14-10 h light-dark cycle at 28.5 °C. Two primers to discriminate the 4-bp size difference between wild type (80 bp) and *dbh^ct806^* mutation (84 bp) were 5′-TACACCATGCTGGAGCATCCC-3′ (forward) and 5′-AGGACTCGCTGGACGCCA-3′ (reverse). Five females of ~3 months of age after gender was confirmed by the genital papilla were housed together in a tank and were subjected to starvation. Approximately 50 females of *dbh^−/−^* and 30 females of *dbh^+/−^* were tested.

## Oogenesis observation

Caenorhabditis. The gonads of *C. elegans* and *C. remanei* were collected after dissection of the head of an animal with a needle. The gonads were fixed in ice-cold methanol for 15 min at −20 °C, washed 3–5 times with PBST (1 × PBS, pH7.4, 0.1% Tween-20 (Sigma, P9416)) and stained with DAPI (1 μg/ml in PBST) for 5 min at room temperature. Approximately 50 animals were stained per strain, and the experiment was repeated at least twice. Ovulation rate were determined as the total of ovulated unfertilized oocytes in the uterus on the plate at 10 h and at 24 h after the late L4 stage at room temperature (approximately at 23 °C)[23]. Females containing quiescent oocytes were mated with wild-type (N2) males at 20 °C. To calculate % unhatched embryo for *fog-2* and *fog-2; tbh-1* (Fig. 4g), the mated females were removed, and the number of unhatched embryos were counted after 24 h. % unhatched embryos = Number of unhatched embryos/total laid eggs.

D. melanogaster. CS media (CSY media without yeast) were used for the protein starvation condition and CSY media for the control condition (moderately protein rich). Ten virgin females (collected within 4 h of eclosion) identified by light body color with greenish substance inside abdomen were transferred to a new CS media and reared at 25 °C for the indicated period of times of 0, 1, 1.5, 2 dpe (day post eclosion). The rearing chamber was maintained at 25 °C with 40% humidity on a 12 h:12 h light-dark cycle. For starvation, 0.8% agar was used. OA (5 mg/ml or 10 mg/ml) was added while making CS media or agar.

After the indicated period, the ovaries were dissected, submerged in halocarbon oil 700 (Sigma, H8898), and observed under the microscope. The stage of an egg chamber was identified based on the described characteristics[44]; the stage 8 egg chambers were identified by accumulation of vitellogenin and germline encircling follicle cells and the stage 6 and 7 egg chambers were identified by the size and the shape[44].

Danio rerio. Five-month-old female fish were starved for 3–6 weeks. After two weeks, they were moved to clean tanks. The ovary sample was prepared as described[69]. Briefly, the ovary was fixed with 4% paraformaldehyde (Sigma, P6148), embedded in a plastic casting, sectioned in 4 μm thickness using a microtome (Leica RM2125) and then stained with 1% neutral red. Eight sections from ovaries of three *dbh^+/−^* and ten sections from ovaries of four *dbh^−/−^* females were selected to count the number of oocytes at the indicated stage. The selected sections were separated more than 40 μm from each other.

## Imaging

*For GFP expression.* GFP-expressing strains of each animal species were mounted and observed under a fluorescence microscope (Zeiss Axio A2 Imager) or a confocal microscope (Olympus) for zebrafish.

The stained gonads of *Caenorhabditis* were observed under a DAPI filter and the differential interference contrast (DIC) setting using a Zeiss Axio A2 Imager with either 40× or 63× objectives. The ovaries of flies were observed using a light-field microscopy at 10× and the ovaries of zebrafish at 5× or 3.2× magnifications. Images were acquired using Zeiss Axiovision software.

For imaging of the ovulation process a live *C. elegans* was observed using a 10× lens at 0 h and 12 h time points. The images of the same animals were taken using a 63× objective at the 21 h time point.

## DNA construct

P$_{ceh-18}$::ser-3. The construct containing the 2.1 kb upstream region of *ceh-18* gene was made and fused to the other construct containing the coding region and 1 kb downstream region of *ser-3* gene. The primers for the former are 5′-ACACATCAGCTTACCTGGCG-3′ (forward) and 5′- TCTCATCCATTCCATATTTCTTGCATTGATGATATGTG-3′ (reverse). The primers for the latter are 5′-AATGCAAGAAATATGGAATGGATGAGAAATACGTTAAAC-3′ (P$_{ceh-18}$::ser-3 joining) and 5′-GAAGGCAGCATGTGTGCAG-3′. The primers to join the two products were 5′-ATTTCACATCAACACGAGTGTACG-3′ (forward) and 5′-CTAAAGTTCAGTTCGTTCACGGAG-3' (reverse).

## RNA interference

Bacteria-mediated feeding RNA interference (RNAi) was performed using RNAi clones from Ahringer's library. For RNAi by injection, double-stranded *C. elegans* tdc-1 and tph-1 RNAs were purified and injected (with 200 ng/ml) into the gonads of females of *C. remanei* at the L4 or young adult stages. Then, the progeny (F$_1$) were scored for the Emo phenotype.

## Brood size

A brood size of each strain of *C. elegans* was measured by transferring a single L4 hermaphrodite to *E. coli*-seeded plates to a new plate every day. The progeny from all the plates from one hermaphrodite were counted after they grew. About eight to thirteen animals were used for each genotype.

## Rescue of *C. elegans* OA mutants

Synchronized L4 stage *C. elegans* were transferred onto NGM plates with or without OA (octopamine hydrochloride, Sigma, O0250) or NE (L-Norepinephrine hydrochloride, Sigma, 74480). After treatment, the ovaries were dissected and stained with DAPI (Sigma, D9542) to examine the phenotype. DAPI stained oocytes in the gonads were observed either at 62 h or 89 h after hatching or at both time points. For the rescue, the ovaries were observed at 89 h after hatching in *tbh-1* and *tdc-1* mutants.

## Antibody staining of MAPK activation

Dissected gonads from ~10−15 worms were fixed with 3% paraformaldehyde for 15 min at 20 °C, washed three times with 1 × PBST and post-fixed in ice-cold methanol for 5 min at 20 °C. The fixed gonads were washed three times with 1 × PBST solution and blocked for 1 h with 0.5% BSA in 1 × PBST. After blocking, gonads were incubated with anti-MAPK-YT (Sigma, M8159, 1:500 dilution) at 4 °C for 16 h, then incubated with an anti-mouse secondary antibody conjugated with Alexa Fluor 488 (Abcam, ab150113, 1:1,000 dilution) for 1 h at room temperature[42].

## Ovulation recording of *C. elegans*

Young adult worms (0 or 1 embryo included) were anesthetized for 30 min in a buffer solution (M9) with 0.01% tetramisole (Sigma, L9756). Gonadal sheath contraction and ovulation were recorded for 60 min at 20 frames/s using a Flea3 digital camera (Point Gray Research). Contractions were counted at 1 min intervals.

## Statistics

We examined whether all experimental groups were normally distributed using Sturges' class number and a symmetric distribution. Most groups appeared to be normally distributed, but some groups are skewed regardless of the sample size. Most OA/NE-lacking mutant groups showed much bigger variances compared with those of control groups. F test was used to determine equality of variances for two independent groups. Although samples in some groups appeared not to be normally distributed, most comparisons between control and test groups were carried out using a two-sided unpaired *t*-test with unequal (for most comparisons) or equal (for a few comparisons) variance, because sample sizes were relatively big. The data in the Result indicate mean values ± SD. All boxplots show biological variability of each group of samples. Box-and-whisker plots show the median, mean (X), interquartile range (IQR) (between the 25th and 75th). The upper whisker indicates the maxima smaller than 1.5 times IQR plus the third quartile, and the lower whisker indicates the minima larger than 1.5 times IQR minus the first quartile. The statistical tests were performed using Microsoft Excel (Microsoft, USA) and the data analysis program GraphPad Prism 9.1 and 9.2 (GraphPad Software Inc, USA). 'n' values represent biologically independent animals except for Fig. 7f–g. For 7f–g. 'n' value represents the number of independent samples.

**Reporting summary**. Further information on research design is available in the Nature Research Reporting Summary linked to this article.

## Data availability
Source data are provided with this paper.

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

## Acknowledgements

We thank Drs. Monastirioti, Hirsh, Green, Kamikouchi, and Prober as well as the *Caenorhabditis* Genetics Center (supported by NIH grant P40 OD010440) and the Bloomington fly stock for strains. We thank Drs. Avery and Strauss for invaluable discussions, Drs. Gallagher and Wyler for critical reading of this manuscript, Dr. Tanaka, Y. Kazuki, Dr. Clam, A. Danielle, B. Lee, Y. Lee, Dr. Padmanabha, W. Seo, H. Kim, S. Ahn. and N. Suryawinata for technical assistance. We thank the members of the Hibi lab and the Min lab for their help in maintaining animals. This work was supported by Inha University and Basic Science Research Program through the National Research Foundation of Korea funded by the Ministry of Education (2017R1D1A1B03036182) (J.K.), the Korea Institute of Toxicology (KK-2011-03) and National Research Foundation of Korea (NRF) (2021R1F1A1045599) (M.H.), JSPS KAKENHI JP18H02448 and CREST Japan Science and Technology Agency (JST) JPMJCR1753 (M.H.), by Virginia Commonwealth University and Nagoya University (Y-J.Y.).

## Author contributions

Y.Y. conceived the idea and supervised the project. J.K. conceived the idea, performed the experiments for Figs. 1, 3–7. M.H. performed the experiments for Figs. 2 and 3. M.H. performed the experiments for Fig. 7 and supervised the zebrafish study. J.K. and Y.Y. wrote the manuscript with input from M.H. and M.H.

## Competing interests

The authors declare no competing interests.

## Ethics statement

The animal work in this study was approved by the Nagoya University Animal Experiment Committee and was conducted in accordance with the "Regulations on Animal Experiments in Nagoya University" and "Guidelines for Proper Conduct of Animal Experiments (Science Council of Japan)".
