## [Peer Review File · Nature Communications]

REVIEWER COMMENTS

Reviewer #1 (Remarks to the Author):

Summary/Critique

This manuscript from Kim and colleagues reports evidence from three biological systems, nematodes, *Drosophila*, and zebrafish suggesting that octopamine signals from cells of the somatic gonad help maintain oocyte quiescence. A strength of the manuscript is the investigation in multiple species. This is also a weakness because the experiments are very preliminary, somewhat superficial, and largely descriptive. The studies do not leverage the wealth genetic, cell biological, and mechanistic information and tools available from longstanding studies of oogenesis in these systems. This is particularly true of the studies in *Drosophila* and zebrafish. The *C. elegans* studies are better and might stand alone with additional clarification and better supporting evidence as discussed in Major Points below.

Major Points

1. The *Drosophila* and zebrafish studies would benefit from molecular endpoints and better integration with the literature in these fields.
2. Regarding the *C. elegans* studies, instead of using RNAi, the authors should construct double mutants between strong loss-of-function mutations in *fog-2* (e.g. *oz40*) and null alleles of *tbh-1* and *tdc-1*. The concern is that the authors will observe that oocytes stack up in the germline and fail to undergo spontaneous meiotic maturation. The reviewer is aware that others have made these strains and obtained that result.
3. The authors should not measure the formation of endomitotic oocytes as an endpoint, but should directly measure rates of meiotic maturation and ovulation. A concern is that the authors are looking at subtle effects on the spontaneous meiotic maturation rate at late time points.
4. The studies should employ molecular readouts of meiotic maturation signaling and better attempt to fit *tbh-1* and *tdc-1* into a genetic pathway. What happens when double mutants are made with genes required in the gonadal sheath cells for oocyte meiotic maturation (e.g., *gsa-1* and *acy-4*).
5. The authors model isn't clear because both *tdc-1* and *tbh-1* are expressed in gonadal sheath cells in both the presence and absence of sperm.
6. The discussion of the mammalian system in the Introduction could be improved by consulting an expert in mammalian oogenesis.

Reviewer #2 (Remarks to the Author):

The authors investigate effects of reducing or eliminating octopamine (OA) signaling on *Drosophila* and nematode oogenesis, and inhibiting noradrenalin (NA) signaling in zebrafish. In all cases they report a marked consequence of releasing oocyte development or activation that is otherwise suppressed by conditions unfavorable for reproduction, while reproduction under optimal conditions is largely unaffected. The authors therefore infer a general role for OA/NA in preserving oocyte "quiescence" under unfavorable conditions.

Here I consider the quality of the evidence and how the results fit into a broader picture.

For *C. elegans* evidence comes from altering two enzymes required for OA synthesis and a receptor identified here as relevant. Phenotypes included mitotic division of haploid oocytes after sperm exhaustion in hermaphrodites and in genetically feminized animals that produce no sperm. A similar phenotype was seen in females of *C. remanei* where there are natural females.

A few shortcomings and questions are likely answered relatively easily:

Are Emo phenotypes seen for older hermaphrodites and females also when males are present?

What explanation might be offered for the limited rescue by OA (Fig. 2c)?

What assurances can be offered that *tbh-1* and *tdc-1* dysfunction affect only OA synthesis?

Most of the data in Fig. 3 lacks clear quantitation, including the proportion of oocytes with Emo phenotypes in feminizing tests. The penetrance of the latter phenotype appears to be quite low in *fog2*; *tbh1* mutants relative to other situations (and no DAPI is shown for these). Any explanations?

Are feminized *C. elegans* with OA defects sterile, like *C. remanei* and is there a significant difference in the timing (onset) of Emo responses presumed to cause sterility in *C. remanei*?

Rescue with SER-3 using a *ceh-18* promoter is used to suggest gonadal sheath cells as the cells receiving OA signals.

Are there additional cells that express *ceh-18* and might plausibly be important?

Are gonadal sheath cells important for any part of the flux of oocytes contributing to oogenesis, fertilization or egg-laying, potentially co-ordinating such roles with influencing oocyte quiescence?

In *Drosophila* the major phenotype reported is the progression of oogenesis beyond stage 8 in young adults starved for protein for *tbh* mutants and an identified receptor, *octb2R*. A strong aspect of these results is the robust rescue of the former, but not the latter mutant phenotype by provision of OA (Fig. 5c).

There is a notable weakness or curiosity to be explained better.

Protein starvation elicits arrest prior to stage 8 and degeneration of stage 8 egg chambers, and the authors state that if degeneration is blocked that maturation continues. I am not sure of the origin of this statement (it is not referenced) but it suggests that regulation of progression beyond stage 8 is entirely dependent on a decision to initiate degeneration or not. Yet the authors main thesis is (or appears to be) that interfering with OA signaling does not affect degeneration but does promote progression. The authors should be clearer about the context of current understanding and its origins, and whether they are proposing that current understanding is incorrect and that there is in fact an OA-enforced growth regulation beyond stage 8.

If the main thesis is instead that arrest prior to stage 8 is absent in OA-deficient animals then I think the phenotype of earlier egg chambers should be reported quantitatively with respect to characteristic P granules and microtubules. Indeed, it is in any case a central issue to establish which starvation-sensitive checkpoints are or are not affected by OA (that includes effects on GSC division).

Currently, the focus of OA action and supporting evidence is not clear to me.

With regard to data, I think the various presentations in Fig. 4b-e are potentially confusing. I presume that b, e, f are per fly and that, because stages 2-7 might be evenly represented and are aggregated in b, the average frequency of each of those stages in most genotypes shown is about 14-15.

Assuming that d-f show non-degenerating egg chambers, the frequency of intact stage 8 chambers seems remarkably high (for controls, given that both prior arrest and degeneration should reduce their frequency). Does stage 8 degeneration initiate only after a significant delay, at least for some egg chambers?

When degeneration is explicitly measured and reported in Fig. 5d the age of flies is not given but the big difference between mutant and control st. 8 intact egg chambers is unexpected because it was not apparent in Fig. 4. Also, if these data are accurate, then the proportion of stage 8 egg chambers undergoing degeneration is much lower for mutants than controls, indicating a defect in degeneration. Indeed, it is hard to understand how so many egg chambers proceed further unless they are spared from degeneration.

Also, the authors note that the investigated OA receptor is expressed in nurse cells but that remains a

speculation regarding site of action since that was not investigated experimentally. It would also be helpful to know whether the allele used (and the tbh allele) is known to be a null mutation or its molecular basis.

The zebrafish studies use one mutant gene (and allele) and one phenotypic measure, and are therefore more limited with regard to site of action, receptor or assurances that only NE action (in a relevant location) is being disrupted.

The broader context of the study as a whole really concerns what was known before, what is now understood better and what remains unresolved.

Here I am being guided largely by the authors' narrative. Mostly, I found the narrative to be informative, especially given the challenge of describing relevant background for three experimental organisms.

One sticking point for me, which is perhaps most readily phrased for the *Drosophila* studies, is whether the authors are suggesting or concluding that there is an active response to protein deficiency (as opposed to the absence of a response to protein sufficiency). The authors say it is not known whether the absence of insulin and JH signals is sufficient to maintain quiescent ovaries or another inhibitory signal is necessary. This needs discussion or refinement as my understanding is that several (maybe all?) facets of quiescence are observed in the absence of those signals (and in some cases are reversed by addition). Later (p12) the idea is repeated, suggesting that there is indeed also a nutrient-responsive neuronal signal, followed by a chain of conditional sentences about expected scenarios.

Do the authors have any evidence that OA presentation to relevant cells is altered in response to nutrition?

Also, have the authors tested whether OA signaling disruption alters insulin peptide secretion or responses or whether one type of signaling overrides the other (under genetic conditions that permit such tests)?

It appears to me that the idea of a nutrient-responsive signal is at this stage speculative and that it is not yet clear whether OA responses act independently of insulin-like signals.

Another significant unknown (in each setting, possibly with the exception of *C. elegans*) appears to be the exact cells responsible for receiving the OA signal, how they then relay the signal and what might be the significance of those choices (why they are suited to the purpose). The evidence in Fig. 1 does not really answer these questions and I am particularly unconvinced from the images about the statement that the stained processes reach germarial regions or why that may be relevant for the reported phenotypes.

Another issue that seems debatable is what is the significance of finding somewhat related roles for OA and NE in three organisms. Much about how OA acts and in which cells remains to be discovered and the manifestations of failures of quiescence and unfavorable conditions examined are quite diverse. It is therefore hard to assess what might be the evolutionary mechanisms and potential common advantages of employing OA/NA.

Altogether, I expect that the authors will be able to resolve most of the issues I have raised with more careful explanations and making available quantitative data more explicit. While I am not personally convinced that investigation of three systems makes the conclusions especially noteworthy, it does seem that there are robust data for more than one system supporting some new insights into an important area of biology that will likely not be very familiar to most readers but nonetheless interesting and informative.

Reviewer #3 (Remarks to the Author):

Remarkably, it is shown in this manuscript that Octopamine/Norepinephrin regulates oogenesis quiescence during starvation conditions in four species, from *C.elegans*, *Drosophila*, to zebrafish. It is well known that oogenesis undergoes quiescence during starvation to save nutrients for survival. However, how the quiescence is mediated molecularly was not known. In *C.elegans* and *Drosophila*, the authors found that oocytes progressed in oogenesis in starvation or protein-deficient conditions when OA (Octopamine) fails to be produced, unlike wild type, which arrest in their development at previtellogenic stages. Importantly, exogenous OA could rescue the defect, demonstrating the specific role of OA in mediating this quiescence. In both *C.elegans* and *Drosophila*, the authors also identified the receptor of OA, which exhibited the same phenotype as the OA-deficient animals. In zebrafish, they show that lack of norepinephrine (OA in vertebrates) causes the same defect that oogenesis proceeds even during starvation conditions. The data strongly support the conclusions.

The authors hypothesize that it is the balance of OA to nutrient signals (Insulin related) that mediates either quiescence or oogenesis progression. It would add important depth to this study, if it were possible to test this in one of the organisms. In *Drosophila* and *C.elegans*, the nutrient signaling pathway is well studied, so it wouldn't be too difficult to test this model.

Minor points:

Fig 2c and Fig 3, please indicate which comparisons are significantly different based on the statistical test performed on the graph itself.

Fig 6d, e. Explain the difference between the black and red scale bars.

Point-by-point response to the reviewers' comments

Dear Editor,

We deeply appreciate the constructive comments from the reviewers. We believe that we addressed all the concerns and that the revised manuscript is improved due to those comments.

The summary of the changes:

- One of the major concerns was how OA signal fits into the known oocyte maturation process in *C. elegans* (i.e., how OA interacts with the sperm signal that controls oocyte maturation). To address this issue, we added new results from three additional experiments, where we examined MAPK activation in OA mutants (now Fig. 2c), constructed the double mutants of *fog-2*; *tbh-1*, and determined the genetic interaction between the sperm signal and OA by measuring ovulation rates and scoring the Emo phenotype (now Fig. 3c and 3d). We also confirmed the double mutants did not maintain quiescent oocytes even when they contained stacked oocytes, by mating the double mutants with males and showing that the oocytes become Emo (now Fig. 4). We confirmed that the defect is only in maintaining quiescence in the absence of sperm; when we mated the double mutant at the L4 stage so that sperm are available before the first oocyte matures, the double mutant produced intact embryos without producing Emo (now Fig. 4f).
- Another major concern was whether OA signal could be an active signal to override the nutrient signals. We added new results from experiments in *D. melanogaster* where we examined how OA and nutrient signals interacted using different combinations of OA and nutrients (now Fig 6e-g). When we treated flies with OA and nutrients varying the concentration of each, OA and nutrients balance each other, indicating that OA is an active signal and thus supporting our hypothesis; A medium concentration of OA (5 mg/ml) inhibits a low level of nutrient signaling (1% yeast). This OA effect is overridden by a high level of nutrient signaling (2.5% yeast). A high concentration of OA (10 mg/ml) inhibits a high level of nutrient signaling (2.5% yeast).
- We revised the text and removed the part regarding egg chamber degradation in *Drosophila* to clarify ambiguity and avoid distraction. We removed the part because the idea that non-degenerated egg chambers would become matured egg chambers was indeed a baseless assumption, and therefore the increased number of matured egg chambers in OA mutants is not due to failure of degeneration. Indeed, even if there are variations in numbers, we always observe a significant number of degenerating egg chambers in OA mutants, showing that the degeneration process does occur. Most importantly, our new results show that the stage 8 egg chambers of the OA mutants actively undergo vitellogenesis whereas the stage 8 egg chambers of the control barely do. If the increased number of vitellogenic egg chambers in OA mutants were due to a defect in degeneration, the ovaries of OA mutants should contain an increased number of stage 8 egg chambers with the same appearance of that of the control (pale with reduced vitellogenesis). This difference clearly indicates that the increased number of matured egg chambers in OA mutants is not due to defects in degeneration but due to failure in remaining quiescent. Therefore, regardless of any concerns about degeneration, our conclusion remains valid especially after our newly added experiments (now Fig. 6e-g).

The reviewer's comments are shown in blue and our responses are in black.

Reviewer #1 (Remarks to the Author):

Summary/Critique

This manuscript from Kim and colleagues reports evidence from three biological systems, nematodes, *Drosophila*, and zebrafish suggesting that octopamine signals from cells of the somatic gonad help maintain oocyte quiescence. A strength of the manuscript is the investigation in multiple species. This is also a weakness because the experiments are very preliminary, somewhat superficial, and largely descriptive. The studies do not leverage the wealth genetic, cell biological, and mechanistic information and tools available from longstanding studies of oogenesis in these systems. This is particularly true of the studies in *Drosophila* and zebrafish. The *C. elegans* studies are better and might stand alone with additional clarification and better supporting evidence as discussed in Major Points below.

Major Points

1. The *Drosophila* and zebrafish studies would benefit from molecular endpoints and better integration with the literature in these fields.

We appreciate the reviewer's comment, and revised the introduction by adding oogenesis of zebrafish and discussing common molecular pathways. However, because there is not much known about 'quiescence' signaling especially in zebrafish, and also because there are many diverse mechanisms due to the big differences among the three models' reproductive strategies and environments, it is not easy to integrate the fields in terms of molecules. In addition, oocyte maturation, especially in response to endocrine signaling, has been intensively studied by many excellent groups. Our discovery suggests that OA/NE acts prior to those signals as a safeguard to protect the oocyte pool when the environment is harsh and unpredictable. We hope the reviewer understands our effort to focus on what is unknown related to our discovery instead of reviewing the vast amount of previously reported findings.

2. Regarding the *C. elegans* studies, instead of using RNAi, the authors should construct double mutants between strong loss-of-function mutations in *fog-2* (e.g. *oz40*) and null alleles of *tbh-1* and *tdc-1*. The concern is that the authors will observe that oocytes stack up in the germline and fail to undergo spontaneous meiotic maturation. The reviewer is aware that others have made these strains and obtained that result.

We agreed and generated the double mutant *fog-2(q71); tbh-1(n3247)*. The double mutants showed the same phenotype as that of the RNAi in first 10 hours from late L4s; they failed to contain quiescent oocytes and ovulated at a faster rate than control (Fig. 3c). We also noticed, as the reviewer mentioned, about 40% of the double mutants did contain stacked oocytes whereas, the remaining 60% did not. However, when we mated them with wild-type males, the stacked oocytes became Emo or produced unhatched embryos, indicating the stacked oocytes were defective and not quiescent (Fig. 4c-f).

The new results are described as:

"Consistent with the RNAi results, *fog-2; tbh-1* double mutants failed to maintain quiescent oocytes from as young as 10 h after L4, and the oocytes became round instead of remaining cylinder-shaped as in *fog-2* single mutants (Fig. 3c). *fog-2; tbh-1* double mutants laid oocytes at a higher ovulation rate than *fog-2* single mutants (Fig. 3d) and became Emo at 1 d (19% Emo, n = 31), 2 d (50% Emo, n

= 24) and 3 d (67% Emo, n = 24) after L4. In contrast, *fog-2* females maintain quiescent oocytes and produce no Emo for 3 days (0% Emo, n = 35).”

and

“We observed that *fog-2; tbh-1* double mutants occasionally stacked oocytes in their gonad arms. To determine whether those stacked oocytes are quiescent and intact, we selected 1-d old females that contained stacked oocytes (42%, n = 31) and mated them with wild-type males. Mated *fog-2; tbh-1* mutants produced the Emo phenotype within 5 h of mating (76.5% Emo, n = 17), whereas mated *fog-2* females or OA-treated *fog-2; tbh-1* did not (0% Emo, n = 13) (Fig. 4c-e).”

3. The authors should not measure the formation of endomitotic oocytes as an endpoint, but should directly measure rates of meiotic maturation and ovulation. A concern is that the authors are looking at subtle effects on the spontaneous meiotic maturation rate at late time points.

We agreed on the concern and measured the rates of meiotic maturation by counting ovulated unfertilized oocytes. The new data are in Fig. 3d. Specifically, to avoid any sampling errors, we measured oocyte maturation rate within 1 d from L4 before over 50% of the females carry Emo (females carrying Emo cannot lay eggs). The oocyte maturation rate of *fog-2; tbh-1* double mutant females is approximately 5 times higher than that of *fog-2* control females. We think this is an obvious phenotype (not subtle) as a similar fold change was reported when *ceh-18(mg57)* mutant females were compared to *fog-2(q71)* or *fog-3(q443)* in the previous study (Govindan et al. Current Biol. 16, 1257-1266, 2006).

4. The studies should employ molecular readouts of meiotic maturation signaling and better attempt to fit *tbh-1* and *tdc-1* into a genetic pathway. What happens when double mutants are made with genes required in the gonadal sheath cells for oocyte meiotic maturation (e.g., *gsa-1* and *acy-4*).

With all due respect, although we appreciate the reviewer’s insight and suggestion, and tested *gsa-1* interaction with OA signal, we do not think the OA signal directly interacts with the sperm signal. Rather, our study suggests the two events of ‘awakening’ and ‘sperm signal-mediated maturation’ are sequential and independent, sharing aspects of oocyte maturation process in all other male/female animals, where oocyte awakening happens before the oocyte meets sperm. As a hermaphrodite, *C. elegans* does not need the 2nd meiotic arrest, the waiting step for sperm, and thus the two events of ‘awakening’ and ‘sperm signal-mediated maturation’ look connected. However, none of OA mutants show any defects in oocyte maturation in the presence of sperm, and the double mutant of *fog-2; tbh-1* only produces Emo when sperm are unavailable, or when sperm are introduced at a late adult stage (Fig. 4f). This indicates that the OA signal is not necessary for ‘sperm signal-mediated maturation’ step. Indeed, our result (Fig. 4) suggests the two signaling pathways are independent.

Specifically, we treated *fog-2(q71)* mutants and *fog-2(q71); tbh-1(n3247)* double mutants with *gsa-1* RNAi and measured oocyte maturation rate. The oocyte maturation rate of the *fog-2(q71)* mutants is comparable to that of *fog-2(q71); tbh-1(n3247)* mutant (0.44 vs 0.53 arm⁻¹ h⁻¹, respectively), suggesting that the G_{so} signal is not epistatic to OA signal. On the contrary, G_{so} is epistatic to the known inhibitory signal G_{oa} (Govindan et al, 2006).

The new result was described as:

“Next, we examined the genetic interaction between OA and sperm signaling by comparing the ovulation rates between *fog-2; tbh-1* and *fog-2* under the condition of reduced sperm signaling of $G_{\alpha s}$, which is encoded by *gsa-1*. We reasoned that if OA directly interacts with sperm signaling, the *gsa-1* phenotype would be epistatic to that of *tbh-1* in maintaining quiescent oocytes; the phenotype of *gsa-1*RNAi would be the same as that of *gsa-1*RNAi in the *tbh-1* background. Because $G_{\alpha s}$ is also required for spermatheca contraction, the oocytes of *gsa-1* RNAi-treated *fog-2; tbh-1* animals became trapped in the spermatheca, where they matured and became Emo in the gonadal sheath, whereas the oocytes of *fog-2* single mutant did not (Fig. 4a, b). Although it is unclear why oocytes of *fog-2* mutants were able to pass spermatheca in *gsa-1* RNAi treatment, the fact that *gsa-1* RNAi produces different phenotypes between *fog-2* and *fog-2; tbh-1* indicates *gsa-1* is not epistatic to *tbh-1*.”

5. The authors model isn't clear because both *tdc-1* and *tbh-1* are expressed in gonadal sheath cells in both the presence and absence of sperm.

We apologize, but this concern is not clear to us. If the reviewer suggests OA production has to be induced only in the absence of sperm, then we think OA should be present as a default signal to maintain quiescence oocytes. We are not aware of signaling pathways that allow the *C. elegans* hermaphrodite to gauge the sperm level and to control their OA production. Our new experiment where we show *fog-2; tbh-1* mutants produced Emo faster after mating also suggests that maintaining quiescent oocytes before meeting sperm is critical.

6. The discussion of the mammalian system in the Introduction could be improved by consulting an expert in mammalian oogenesis.

We really thank the reviewer for this comment. We have revised the Introduction.

Reviewer #2 (Remarks to the Author):

The authors investigate effects of reducing or eliminating octopamine (OA) signaling on *Drosophila* and nematode oogenesis, and inhibiting noradrenalin (NA) signaling in zebrafish. In all cases they report a marked consequence of releasing oocyte development or activation that is otherwise suppressed by conditions unfavorable for reproduction, while reproduction under optimal conditions is largely unaffected. The authors therefore infer a general role for OA/NA in preserving oocyte “quiescence” under unfavorable conditions.

Here I consider the quality of the evidence and how the results fit into a broader picture.

For *C. elegans* evidence comes from altering two enzymes required for OA synthesis and a receptor identified here as relevant. Phenotypes included mitotic division of haploid oocytes after sperm exhaustion in hermaphrodites and in genetically feminized animals that produce no sperm. A similar phenotype was seen in females of *C. remanei* where there are natural females.

A few shortcomings and questions are likely answered relatively easily:

1. Are Emo phenotypes seen for older hermaphrodites and females also when males are present?

We tested this idea in a feminized mutant (*fog-2*) and *C. remanei* females. The current understanding is that constant presence of sperm signal prevents Emo by coordinating oocyte maturation and ovulation. Consistent with this understanding, both the *fog-2 C. elegans* and female *C. remanei* without OA did not produce Emo phenotype if they are mated from the final larval (L4) stage – and thus guarantee the constant sperm signal. In this case, 3-d old females did not produce Emo. However, they produce Emo if males are present after the female has been an adult for at least one day. The *fog-2; tbh* data are presented in Fig. 4f and *C. remanei* data are mentioned in the text.

2. What explanation might be offered for the limited rescue by OA (Fig. 2c)?

We appreciate this comment. We found the original graph included (inadvertently) all the time points including several earlier timepoints we used to observe the time course. When we corrected this error and plotted the graph with the data from 89 h after hatch, as written in the figure legend, the analysis showed that exogenous OA rescues Emo phenotype in both *tbh-1* and *tdc-1* mutants nearly completely (Fig. 2d). The full rescue on Emo phenotype by OA is seen repeatedly in other backgrounds (Fig. 3c, d and Fig. 4e, g).

3. What assurances can be offered that *tbh-1* and *tdc-1* dysfunction affect only OA synthesis?

It is our understanding that as these are enzymes to process octopamine and tyramine, and we cannot think of any other biological processes these mutants would affect. If the concern is about potential tyramine function, our experiments adding octopamine that rescued all of the phenotypes that we examined suggests that the oocyte phenotype is due to lack of octopamine.

4. Most of the data in Fig. 3 lacks clear quantitation, including the proportion of oocytes with Emo phenotypes in feminizing tests. The penetrance of the latter phenotype appears to be quite low in *fog2; tbh1* mutants relative to other situations (and no DAPI is shown for these). Any explanations?

Thank you. To address this and the reviewer #1's concerns, we generated a feminized mutant lacking OA (*fog-2; tbh-1*). This allows us to precisely quantify relevant phenotypes including Emo (Fig. 3c-e). In summary, (a) we measured an oocyte maturation rate in the *fog-2; tbh-1* mutant

females and found it is higher than *fog-2* mutant females (Fig. 3c-d). (b) we quantified the fraction of Emo, stacked oocytes, and awoken oocytes in the *fog-2; tbh-1* females and mated *fog-2; tbh-1* mutants and found most of stacked oocytes (even before they show apparent Emo) are not functional (Fig. 3c-e; Fig. 4a-g).

Although DAPI is better to examine large number of animals for Emo phenotype, it cannot be used to examine live animals (since it requires fixation) and thus cannot provide high-resolution DIC images necessary to observe the oocyte maturation process.

5. Are feminized *C. elegans* with OA defects sterile, like *C. remanei* and is there a significant difference in the timing (onset) of Emo responses presumed to cause sterility in *C. remanei*?

Thank you for the insightful question. While we are addressing the question of, “Are Emo phenotypes seen for older hermaphrodites and females also when males are present?”, we also address the timing issue. Specifically, (a) when *fog-2; tbh-1* females are mated with males from late L4 so that sperm will be present as soon as they become young adults, they are completely normal and fertile (Fig. 4f). (b) However, when 1-d old *fog-2; tbh-1* females were mated with wild type males, 76% of females produced Emo. These data implicate that even if a certain fraction of 1-d old *fog-2; tbh-1* females accumulate oocytes, those stacked oocytes are not quiescent, and thus cannot undergo normal oocyte maturation, resulting in Emo (Fig. 4c-d).

6. Rescue with SER-3 using a *ceh-18* promoter is used to suggest gonadal sheath cells as the cells receiving OA signals. Are there additional cells that express *ceh-18* and might plausibly be important?

Thank you. We clarified, “SER-3 is expressed in head muscles, several neurons, intestine, spermatheca and gonadal sheath cells. To locate the SER-3 action, we targeted *ser-3* expression to gonadal sheath cells using a *ceh-18* promoter. CEH-18 is a Pit-1/Oct-1,2/Unc-86 (POU) domain-containing transcription factor required for gonadal sheath cell differentiation. Although CEH-18 is broadly expressed including in muscles, neurons, and the gonads⁴¹, in the gonad, it is only expressed in gonadal sheath cells and not detected in sperm or oocytes. In addition, the only tissues where both *ceh-18* and *ser-3* are expressed are gonadal sheath cells and spermatheca. This construct rescues the Emo phenotype in *ser-3* mutants (Fig. 2e, *ser-3* TG). Thus, we suggest SER-3 function in gonadal sheath cells is sufficient to maintain quiescent oocytes.”

7. Are gonadal sheath cells important for any part of the flux of oocytes contributing to oogenesis, fertilization or egg-laying, potentially co-ordinating such roles with influencing oocyte quiescence?

Thank you for an insightful and important question. It is known that the gonadal sheath cells communicate with the oocytes via gap junctions and are involved in ovulation by mechanically pushing oocytes toward to the spermatheca. However, mutants simply causing slow contraction (i.e. *egl-30* which encodes $G_{\alpha q}$) do not produce Emo, although they lay significantly fewer progeny than wild-type (Govindan *et al.*, Development 136, 2211-2221, 2009; Brundage *et al.*, Neuron, 16, 999-1009, 1996). Considering that the wild-type and *egl-30* hermaphrodites have the same amount of sperm, and that reduced sheath contraction itself does not produce Emo, it is implicated that oocyte maturation and the rate of the gonadal sheath contraction are well-coordinated in these mutants. The detailed mechanism especially whether or how any such coordination would affect oocyte quiescence in specific, however, is not well understood. Currently, we do not feel it is the focus of our manuscript and hope to address it in the future.

8. In *Drosophila* the major phenotype reported is the progression of oogenesis beyond stage 8 in young adults starved for protein for *tbh* mutants and an identified receptor, *octb2R*. A strong aspect of these results is the robust rescue of the former, but not the latter mutant phenotype by provision of OA (Fig. 5c). There is a notable weakness or curiosity to be explained better.

Protein starvation elicits arrest prior to stage 8 and degeneration of stage 8 egg chambers, and the authors state that if degeneration is blocked that maturation continues. I am not sure of the origin of this statement (it is not referenced) but it suggests that regulation of progression beyond stage 8 is entirely dependent on a decision to initiate degeneration or not. Yet the authors main thesis is (or appears to be) that interfering with OA signaling does not affect degeneration but does promote progression. The authors should be clearer about the context of current understanding and its origins, and whether they are proposing that current understanding is incorrect and that there is in fact an OA-enforced growth regulation beyond stage 8.

Thank you for pointing out this important issue. We do not suggest a novel mechanism of OA in the progression beyond stage 8 egg chambers, or its role in degeneration. Please see our response to the next.

If the main thesis is instead that arrest prior to stage 8 is absent in OA-deficient animals then I think the phenotype of earlier egg chambers should be reported quantitatively with respect to characteristic P granules and microtubules.

The characteristic P granules and microtubules under protein-starvation condition were mainly observed in stage 6/7 egg chambers (Shimada et al, 2011, Burn et al, 2015). However, as shown in the images below and explained next, stage 7 egg chambers were rarely seen in the *tbh*^{-/-} mutant ovaries under starvation. This impedes any comparison of the distribution of the P granule marker (Ypsilon schachtel: Yps) and Tubulin between the controls and *tbh*^{-/-} mutants.

Indeed, it is in any case a central issue to establish which starvation-sensitive checkpoints are or are not affected by OA (that includes effects on GSC division). Currently, the focus of OA action and supporting evidence is not clear to me.

The currently known starvation check points are GSC and stage 6/7 egg chambers. Because it takes approximately 7 days from a germ line stem cell (GSC) to develop to a stage 2 egg chamber (Spradling, 1993), and because all our OA experiments were done using flies in *2 dpe* (2 days after eclosion), we can safely exclude the possibility that the increased number of vitellogenic egg chambers (~5× more than control in *tbh*^{-/-} mutants, Fig 6b, d) in OA mutants was from any potential increased number of GCS.

We tested a complete starvation condition where we removed all nutrient sources including sugar and cornmeal. This complete starvation blocks almost all egg chambers from awakening in control (Canton-S) of age of *2 dpe* (Fig. 6d, n=15). Most of the egg chambers of Canton-S remained at stage 6/7 or earlier stages, confirming that stage 6/7 is a starvation checkpoint (marked as Q in the below Figure R1, R2). The egg chambers of *tbh*^{-/-} mutants, however, contain fewer stage 6/7 egg chambers but instead many vitellogenic egg chambers including stage 14 egg chambers (Fig. 6d, n=56), confirming the role of OA in maintaining the stage 6/7 egg chambers.

To support our explanation, we provide below images of oogenesis of Canton-S and *tbh*^{-/-} mutants under complete starvation. Because Fig. 6a already shows the clear difference in oogenesis between

Canton-S and *tbh*^{-/-} mutants under the protein starved conditions, we did not add them to the main figure, to avoid repetition.

Figure R1. Difference in distribution of quiescent egg chambers between *Canton-S* and the *tbh*^{-/-} mutant under complete starvation.

Figure R2. Difference in the number of awakened egg chambers between *Canton-S* and the *tbh*^{-/-} mutant under complete starvation.

(Left) The ovary of Canton-S contains five “quiescent stage 7 egg chambers” (Q), which can be determined by the size of stage 7/8 egg chambers and lack of vitellogenesis, and four small vitellogenic egg chambers (*) at 2 *dpe*, suggesting low level of spontaneous vitellogenesis under complete starvation at 2 *dpe*.

(Right) In contrast, the ovary of *tbh*^{-/-} mutants contains eight vitellogenic egg chambers (4 of s14, 2 of s13, 1 of s12, 1 of s10B; some egg chambers are out of the frame), four stage 9 egg chambers (#), four small vitellogenic egg chambers (*) as well as at least 5 degenerating stage 8/9 egg chambers (*), suggesting 5 times more awakened egg chambers (total 21) than that of Canton-S control.

9. With regard to data, I think the various presentations in Fig. 4b-e are potentially confusing. I presume that b, e, f are per fly and that, because stages 2-7 might be evenly represented and are aggregated in b, the average frequency of each of those stages in most genotypes shown is about 14-15.

We clarified in the figure legends as:

“b: The number of previtellogenic egg chambers (stages 2–7) per fly”

“d-f: The distribution of each stage egg chambers in the ovaries of each *tbh*^{+/-} and *tbh*^{-/-} female virgin”

Assuming that d-f show non-degenerating egg chambers, the frequency of intact stage 8 chambers seems remarkably high (for controls, given that both prior arrest and degeneration should reduce their frequency). Does stage 8 degeneration initiate only after a significant delay, at least for some egg chambers?

The reviewer is correct on both accounts. Fig. 4d-f (now Fig. 5d-f) show non-degenerating egg chambers, and although our data are insufficient to clearly address the kinetics of stage-8 degeneration, our observation and estimation suggests a significant delay in degeneration in control. Please see our response to the next concern.

When degeneration is explicitly measured and reported in Fig. 5d the age of flies is not given but the big difference between mutant and control st. 8 intact egg chambers is unexpected because it was not apparent in Fig. 4. Also, if these data are accurate, then the proportion of stage 8 egg chambers undergoing degeneration is much lower for mutants than controls, indicating a defect in degeneration. Indeed, it is hard to understand how so many egg chambers proceed further unless they are spared from degeneration.

For questions regarding Fig. 4 (Fig. 5d-f in the revised manuscript):

Due to technical difficulties to measure kinetics which requires time course of oogenesis ideally in each ovariole, we compared the total number of all countable previtellogenic egg chambers per fly at eclosion at 1 *dpe*, 1.5 *dpe*, and 2 *dpe* between *tbh*^{-/-} and *tbh*^{+/-} mutants to gain snapshots of the progress. Although we understand the reviewer’s concern regarding that showing duplicate data could be confusing, we also feel individual data would be informative.

Regarding degeneration kinetics concerning Fig. 4 and Fig. 5d (we removed):

According to the literature, at eclosion, an ovary contains 16 ovarioles. Each ovariole contains one stage 6/7 previtellogenic egg chamber as the most advanced. It would grow to stage 14 at 1.5 *dpe* in

the presence of nutrients. The calculation is based on the 34.5 h to progress from stage 6 to 14 (Spradling, "Developmental genetics of oogenesis", 1993). Under the starvation conditions, Canton-S flies rarely undergo vitellogenesis; only 4 vitellogenic egg chambers are identifiable at 2 *dpe* (Above image: Left, *). They are all stage 8, suggesting arrest of oogenesis process arrests and delayed degeneration.

In our comparison between *tbh*^{-/-} and *tbh*^{+/-} mutants under protein starvation conditions, the ovaries of each *tbh*^{-/-} fly contains 18 vitellogenic egg chambers in average at 1 *dpe*. The number of vitellogenic egg chambers is the sum of stages 8 to 12 egg chambers. At 1.5 *dpe*, 7.3 stage 14 egg chambers appear in average. Because it takes approximately 0.5 d from stage 10 to stage 14, we estimate almost all vitellogenic egg chambers of stage 10-12 (7 in average) in 1 *dpe* grow to stage 14. This is consistent with the literature that there is no halt or degeneration after stage 10 in growth of egg chamber. In contrast, the difference between the average number of the stage 8/9 egg chambers at 1 *dpe* (11 in average) and the number of stage 10-13 at 1.5 *dpe* (5.7 in average) which were supposed to grow from the stage 8/9 at 1 *dpe*. From these data approximately degeneration of 5.3 egg chambers is deduced (11 stage 8/9 at 1 *dpe* – 5.7 stage 10-13 at 1.5 *dpe* = 5.3 degeneration).

The ovaries of each *tbh*^{+/-} contains 16 vitellogenic egg chambers (sum of stage 8 to 11 egg chambers, no stage 12 appears) in average at 1 *dpe*. At 1.5 *dpe*, no stage 14 egg chambers in average appears. The difference between the average number of the stage 8/9 egg chambers at 1 *dpe* (14 in average) and the number of stage 10-13 at 1.5 *dpe* (5 in average) which were supposed to grow from the stage 8/9 at 1 *dpe*. From these data, approximately degeneration of 9 egg chambers is deduced (14 stages 8/9 at 1 *dpe* – 5 stages 10-13 at 1.5 *dpe* = 9 degeneration).

The above estimation suggests potentially reduced degeneration in *tbh*^{-/-} compared to *tbh*^{+/-} mutants (5.3 vs 9). However, we do not know whether the difference is significant. In theory, the pool of stage 8 egg chambers would be reduced by increase of degeneration and/or by increase of progression to the next stage. When it is impossible to trace a single egg chamber for its oogenesis progress, it is hard to discern the two possibilities. Nonetheless, we clearly see substantial number of degenerating egg chambers in *tbh*^{-/-}, indicating that degeneration process does occur. Most importantly, the stage 7/8 egg chambers of Canton-S appear pale, whereas those of *tbh*^{-/-} appear dark, indicating active vitellogenesis in *tbh*^{-/-} mutants. This difference cannot be explained by failure of degeneration; if so, *tbh*^{-/-} mutants should contain the same pale stage 7/8 seen in Canton-S with an increased number. Instead, the egg chambers in *tbh*^{-/-} mutants actively progress to vitellogenic stages compared to either *tbh*^{+/-} or two controls (Canton-S, *w*¹¹¹⁸). This strongly supports our conclusion that OA is required to maintain quiescent egg chambers of stage 6/7.

Based on this deduction and that the assumption that non-degenerated egg chambers would become matured egg chambers was baseless, we removed Fig. 5d of the original manuscript to avoid distraction.

10. Also, the authors note that the investigated OA receptor is expressed in nurse cells but that remains a speculation regarding site of action since that was not investigated experimentally. It would also be helpful to know whether the allele used (and the *tbh* allele) is known to be a null mutation or its molecular basis.

Thank you. We revised:

(a) “*tbh^{nM18}* a null allele that does not produce OA”, and cited again the paper by Monastirioti M, Linn CE, Jr., White K. Characterization of *Drosophila* tyramine beta-hydroxylase gene and isolation of mutant flies lacking octopamine. *J Neurosci* **16**, 3900-3911 (1996).

(b) “An *in situ* hybridization study shows *octβ2R* expression in the nurse cells of previtellogenic egg chambers, suggesting it might be a receptor for the OA action in oocyte quiescence. The allele *octβ2R⁰⁵⁶⁷⁹* contains a piggyBac insertion and is considered as a significantly reduced-function mutant.” And we added two references.

11. The zebrafish studies use one mutant gene (and allele) and one phenotypic measure, and are therefore more limited with regard to site of action, receptor or assurances that only NE action (in a relevant location) is being disrupted.

We agree. We added sentences to support the use of this mutant, toned down the section regarding NE function in oocyte quiescence in zebrafish, and discussed the limitation in the Discussion as follows, “Although the zebrafish study was based on the observation made from a mutant and we have not identified the receptor(s) and the action site in the ovary, we could observe that the quiescent oocytes were maintained in the control whereas they were not in the mutant, which is the same as in the other animals we observed.” Thank you.

12. The broader context of the study as a whole really concerns what was known before, what is now understood better and what remains unresolved. Here I am being guided largely by the authors’ narrative. Mostly, I found the narrative to be informative, especially given the challenge of describing relevant background for three experimental organisms. One sticking point for me, which is perhaps most readily phrased for the *Drosophila* studies, is whether the authors are suggesting or concluding that there is an active response to protein deficiency (as opposed to the absence of a response to protein sufficiency). The authors say it is not known whether the absence of insulin and JH signals is sufficient to maintain quiescent ovaries or another inhibitory signal is necessary. This needs discussion or refinement as my understanding is that several (maybe all?) facets of quiescence are observed in the absence of those signals (and in some cases are reversed by addition). Later (p12) the idea is repeated, suggesting that there is indeed also a nutrient-responsive neuronal signal, followed by a chain of conditional sentences about expected scenarios. Do the authors have any evidence that OA presentation to relevant cells is altered in response to nutrition? Also, have the authors tested whether OA signaling disruption alters insulin peptide secretion or responses or whether one type of signaling overrides the other (under genetic conditions that permit such tests)? It appears to me that the idea of a nutrient-responsive signal is at this stage speculative and that it is not yet clear whether OA responses act independently of insulin-like signals.

We appreciate the insightful summary/points. The same issue was brought up by reviewer #3. To address this concern, we performed a new set of experiments. When we treated flies with OA and nutrients varying the concentration of each, OA and nutrients balance each other, indicating that OA is an active signal and thus supporting our hypothesis; A medium concentration of OA (5 mg/ml) inhibits low level of nutrient signaling (1% yeast). This OA effect is overridden by high level of nutrient signaling (2.5% yeast). A high concentration of OA (10 mg/ml) inhibits high level of nutrient signaling (2.5% yeast) (Fig. 6 e-g).

13. Another significant unknown (in each setting, possibly with the exception of *C. elegans*) appears to be the exact cells responsible for receiving the OA signal, how they then relay the signal and what might be the significance of those choices (why they are suited to the purpose). The evidence in Fig.

1 does not really answer these questions and I am particularly unconvinced from the images about the statement that the stained processes reach germarial regions or why that may be relevant for the reported phenotypes.

For the concern regarding detailed molecular mechanisms (a significant unknown), please consider our attempt to address the significance issue below (#14). For the expression, we revised the text, "A GAL4 line that drives GFP expression by a *tdc2* promoter showed *tdc2* is expressed in many varicosities or boutons of neurons covering the ovary (Fig. 1c)." and removed 'germaria'.

14. Another issue that seems debatable is what is the significance of finding somewhat related roles for OA and NE in three organisms. Much about how OA acts and in which cells remains to be discovered and the manifestations of failures of quiescence and unfavorable conditions examined are quite diverse. It is therefore hard to assess what might be the evolutionary mechanisms and potential common advantages of employing OA/NA.

We do not attempt to address the detailed molecular mechanisms among all three models. That would require three different papers (we think). Rather, we investigated the molecular and cellular mechanism in *C. elegans*, and asked whether the same molecule serves a conserved function by testing two other models. We believe the focus and merit of our paper is that novel finding among three such distant animals. Regardless of the detailed molecular mechanisms, the observations we made clearly share a common theme: **in the absence of OA or NE, quiescent oocytes are not maintained**. This is the first report of such molecule in any animals and also a first report of OA action in oocyte quiescence. In fact, we found the idea that Nature uses the same molecule to protect oocytes from environmental uncertainty is exciting. It is even more so when considering these animals use completely different reproductive strategies.

Altogether, I expect that the authors will be able to resolve most of the issues I have raised with more careful explanations and making available quantitative data more explicit. While I am not personally convinced that investigation of three systems makes the conclusions especially noteworthy, it does seem that there are robust data for more than one system supporting some new insights into an important area of biology that will likely not be very familiar to most readers but nonetheless interesting and informative.

Thank you so much for the critical reading and identifying the points to clarify. We deeply appreciate it.

Reviewer #3 (Remarks to the Author): Nutrient and OA balance in flies.

Remarkably, it is shown in this manuscript that Octopamine/Norepinephrin regulates oogenesis quiescence during starvation conditions in four species, from *C.elegans*, *Drosophila*, to zebrafish. It is well known that oogenesis undergoes quiescence during starvation to save nutrients for survival. However, how the quiescence is mediated molecularly was not known. In *C.elegans* and *Drosophila*, the authors found that oocytes progressed in oogenesis in starvation or protein-deficient conditions when OA (Octopamine) fails to be produced, unlike wild type, which arrest in their development at previtellogenic stages. Importantly, exogenous OA could rescue the defect, demonstrating the specific role of OA in mediating this quiescence. In both *C.elegans* and *Drosophila*, the authors also identified the receptor of OA, which exhibited the same phenotype as the OA-deficient animals. In zebrafish, they show that lack of norepinephrine (OA in vertebrates) causes the same defect that oogenesis proceeds even during starvation conditions. The data strongly support the conclusions.

The authors hypothesize that it is the balance of OA to nutrient signals (Insulin related) that mediates either quiescence or oogenesis progression. It would add important depth to this study, if it were possible to test this in one of the organisms. In *Drosophila* and *C.elegans*, the nutrient signaling pathway is well studied, so it wouldn't be too difficult to test this model.

We really appreciate this insightful suggestion. After considering how to pursue this idea, we decided to test it directly balancing OA and nutrients instead of using nutrient sensing mutants such as insulin mutants. The main reason is that such mutants exhibit so many diverse phenotypes, including reproduction and metabolism as well as development and growth, in both *C. elegans* and *Drosophila*. In addition to the concern that this pleiotropic phenotype creates technical difficulties in matching the ages and conditions, since those mutants are already adapted to a low nutrient state, we were concerned that the results might not be clear to be conclusively interpreted. The approach we took, however, supports our hypothesis that OA and nutrients balance each other; When we treated flies with OA and nutrients varying the concentration of each other, a medium concentration of OA (5 mg/ml) inhibits low level of nutrient signaling (1% yeast). This OA effect is overridden by high level of nutrient signaling (2.5% yeast). A high concentration of OA (10 mg/ml) inhibits high level of nutrient signaling (2.5% yeast). The new data are added in Fig. 6e and g. We thank the reviewer very much. Adding this piece of evidence helps us to clarify our hypothesis and thus improve the paper.

Minor points:

Fig 2c and Fig 3, please indicate which comparisons are significantly different based on the statistical test performed on the graph itself.

We appreciate the comment and indicated the significance on the graph.

Fig 6d, e. Explain the difference between the black and red scale bars.

Thank you so much for pointing out this error! We fixed it.

REVIEWER COMMENTS

Reviewer #2 (Remarks to the Author):

The authors have responded diligently to various questions raised and the manuscript reads quite smoothly at a superficial level.

However, I still have two sets of comments.

The first concern presentation and are easily addressed.

"Oocyte quiescence" is used repeatedly and as a unifying expression. However, I think that the term will be understood in various ways and many are not consistent with the evidence. One meaning is failure to progress through cell cycles- for *Drosophila* egg chambers progressing past stage 8 that is not an appropriate description. Related to that, Fig. 5 implies that stage 8 might involve a transition from prophase 1 to metaphase 1 (because of ambiguous alignment & contraction of stages). Other sources repeatedly state that occurs at stage 13. Also, this maturation is not measured in any of the studies presented, so there is no evidence related to oocyte cell cycle progression. The authors should make clear that they are not claiming any relevance to the prophase to metaphase transition. It may be that "developmental arrest" is a more appropriate term. This is not trivial in the sense that the *Drosophila* example may not involve events in the oocyte at all.

Order of presentation and specificity is, in my opinion, not optimal in some places:

"lacking OA" line 181 and "OA mutants" line 223 should be replaced or supplemented by the actual genotype (they are not literally OA mutants and they have not been shown here to lack OA, though I have no problem with that as an inference from other studies).

Mutant phenotypes are often presented before control behaviors- line 211, 304 and other places, including behavior of animals without starvation (first).

There are some odd statements:

Line 291 *Drosophila* has 2nd meiotic arrest. So what? It is after all of the regulated events studied here and responds to sperm (should be clearly stated somewhere).

301 quiescent oocytes not easily staged in well-fed flies- there is not any evidence of arrest of any type in well-fed flies is there, so what is to be staged?

The discussion (and a response to my first review) suggest that the authors believe they have shown that an OA signal actively responds to a lack of nutrition. I strongly disagree. To show that would require demonstrating a change in OA production in response to nutrients and/or genetically inactivating such a response mechanism (without eliminating OA production altogether) and seeing a defect in the overall oogenesis response to nutrient deprivation (eliminating other response pathways if necessary). It is not addressed by synthetically applying different amounts of OA and seeing different responses. This is an important unjustified claim. OA could be produced at constant level; if it is not, then the source signal and its relay would be interesting, but none of that has been established here.

The second set of comments concern the *Drosophila* evidence.

The aspect of degeneration of stage 8 oocytes has now been eliminated. Consequently, the text is easy to read and sounds convincing. But when I look at the data in Figs. 5/6 I am far from convinced. In essence, this is because the most striking difference is the accumulation of stage 14 egg chambers

in mutant conditions. However, I presume that this has two components- arrival at stage 14 and loss of stage 14 egg chambers. I understand that the latter is blocked in *tbh* mutants because egg-laying is blocked. That blockage could plausibly account for most of the differences in stage 14 egg chamber accumulation, right? Progress beyond stage 8 manifest by the abundance of stage 9-13 egg chambers is barely evident and not significant. In theory, lack of progress for controls would be evident from an accumulation of stage 8 egg chambers. However, that is not seen at all (Fig. 5e, f). That may be because of degeneration of stage 8 egg chambers, so that measuring degenerating stage 8 egg chambers may be important, if it can be done effectively (unfortunately, I expect it is too transient to help much).

So, overall, looking at the data in Fig. 5 I am not sure I see convincing evidence of significant differences in progress past stage 8. Fig. 5d is the most convincing and perhaps summing all post-stage 8 stages shows significant differences.

The authors know their data better than I do, so perhaps they can explain why my arguments here are incorrect. (If my arguments are valid and Fig. 5d is sufficient, then I think it is important for the authors to explain that accumulation of stage 14 egg chambers has two contributing factors).

If not, then one possible solution may be available through the Oct-b2R mutation. Does this mutation also block egg-laying? If not, then stage 14 accumulation would be due only to a change in influx and not efflux (but I suspect the answer is that egg-laying is blocked and is not rescued by OA).

Another possible solution is to compare the mutant strains with and without good nutrition because the lack of efflux from stage 14 should be the same in both cases. Perhaps the authors already conducted that experiment. Some relevant data are in Fig. 6f. I am suggesting that the mutants in rich medium are used in the first evaluations of whether they affect progress past stage 8.

Reviewer #3 (Remarks to the Author):

The authors have addressed my point of testing directly OA and nutrient availability by examining this in *Drosophila*. The results nicely support their model and strengthen the manuscript. With all the additional results, it is greatly strengthened and I recommend publication.

Reviewer #4 (Remarks to the Author):

This revised manuscript describes an important, general, and convincing finding of octopamine action on oocyte maturation across animal phyla. I see no need for further revisions.

Point-by-point response to the reviewer's comments

Dear Editor,

We believe that we addressed all the concerns and that the revised manuscript is improved due to the comments. We appreciate that.

The reviewer's comments are shown in blue and our responses are in black.

Reviewer #2 (Remarks to the Author):

The authors have responded diligently to various questions raised and the manuscript reads quite smoothly at a superficial level. However, I still have two sets of comments.

1. The first concern presentation and are easily addressed.

"Oocyte quiescence" is used repeatedly and as a unifying expression. However, I think that the term will be understood in various ways and many are not consistent with the evidence. One meaning is failure to progress through cell cycles- for *Drosophila* egg chambers progressing past stage 8 that is not an appropriate description. Related to that, Fig. 5 implies that stage 8 might involve a transition from prophase 1 to metaphase 1 (because of ambiguous alignment & contraction of stages). Other sources repeatedly state that occurs at stage 13. Also, this maturation is not measured in any of the studies presented, so there is no evidence related to oocyte cell cycle progression. The authors should make clear that they are not claiming any relevance to the prophase to metaphase transition. It may be that "developmental arrest" is a more appropriate term. This is not trivial in the sense that the *Drosophila* example may not involve events in the oocyte at all.

Indeed, there is no definition regarding starvation-induced 'oocyte quiescence' in *Drosophila*. However, the starved previtellogenic egg chambers share the characteristics of quiescent oocytes in other animals. Therefore, for consistency of the paper, we clarify our definition of quiescence in the beginning of *Drosophila* section as the reviewer suggested so as to use the term throughout the section.

It is now written:

"This diapause is considered different from reduced oogenesis by starvation, during which the proliferation rate of follicle cells reduces to $\frac{1}{4}$ of that of well-fed condition^{14, 45, 46}. Starved previtellogenic egg chambers share certain defined characteristics of oocyte quiescence in other animal germlines such as redistribution of ribonucleoprotein complex components and cortically condensed microtubules^{46, 47}. Based on this similarity, hereafter we call reduction of oogenesis or temporary developmental arrest of previtellogenic egg chambers induced by starvation as oocyte quiescence."

Order of presentation and specificity is, in my opinion, not optimal in some places:

"lacking OA" line 181 and "OA mutants" line 223 should be replaced or supplemented by the actual genotype (they are not literally OA mutants and they have not been shown here to lack OA, though I have no problem with that as an inference from other studies).

We clarified:

The *tbh-1* mutants carry a deletion covering the 5th and 6th exons and do not produce OA²⁵.

Mutant phenotypes are often presented before control behaviors- line 211, 304 and other places, including behavior of animals without starvation (first).

We disagree on this point. It was intended to emphasize the mutant phenotype. We ask the reviewer to understand that this is rather a writing style than a rule to follow. We appreciate understanding.

There are some odd statements:

Line 291 *Drosophila* has 2nd meiotic arrest. So what? It is after all of the regulated events studied here and responds to sperm (should be clearly stated somewhere).

We agreed and deleted “which, unlike *C. elegans*, exhibits a 2nd meiotic arrest.”

301 quiescent oocytes not easily staged in well-fed flies- there is not any evidence of arrest of any type in well-fed flies is there, so what is to be staged?

We deleted “suggesting quiescent oocytes are not easily staged in well-fed *Drosophila*.”

The discussion (and a response to my first review) suggest that the authors believe they have shown that an OA signal actively responds to a lack of nutrition. I strongly disagree. To show that would require demonstrating a change in OA production in response to nutrients and/or genetically inactivating such a response mechanism (without eliminating OA production altogether) and seeing a defect in the overall oogenesis response to nutrient deprivation (eliminating other response pathways if necessary). It is not addressed by synthetically applying different amounts of OA and seeing different responses. This is an important unjustified claim. OA could be produced at constant level; if it is not, then the source signal and its relay would be interesting, but none of that has been established here.

We could not locate the sentence ‘OA signal **actively responds** to a lack of nutrition.’ in our manuscript.

What we wrote was, “OA **could** play an **active role in balancing** nutrient signaling especially when the level of nutrients is insufficient to maintain quiescent egg chambers.”

It appears that there is misunderstanding in interpretation of ‘an active role’. We used ‘active’ as ‘the signal that could override another signal when it is strong enough’ – as a contrast to a default or a passive signal. And we did show a high concentration of OA could override nutrient signal. This, at least to us, suggests that OA signal is not a default (or passive) which would simply disappear whenever nutrient is present regardless the level of nutrients. Rather, it could override nutrient signal if there is enough amount of octopamine.

To avoid further confusion, we removed ‘active’ and rephrased it.

It is now written:

“Our results that the nutrient signal and OA signal could balance each other suggest that when the level of nutrients is insufficient, OA signal could be critical to maintain quiescent egg chambers.”

2. The second set of comments concern the *Drosophila* evidence.

The aspect of degeneration of stage 8 oocytes has now been eliminated. Consequently, the text is easy to read and sounds convincing. But when I look at the data in Figs. 5/6 I am far from convinced. In essence, this is because the most striking difference is the accumulation of stage 14 egg chambers in mutant conditions. However, I presume that this has two components- arrival at stage 14 and loss of stage 14 egg chambers. I understand that the latter is blocked in *tbh* mutants because egg-laying is blocked. That blockage could plausibly account for most of the differences in stage 14 egg chamber accumulation, right? Progress beyond stage 8 manifest by the abundance of stage 9-13 egg chambers is barely evident and not significant. In theory, lack of progress for controls would be evident from an accumulation of stage 8 egg chambers. However, that is not seen at all (Fig. 5e, f). That may be because of degeneration of stage 8 egg chambers, so that measuring degenerating stage 8 egg chambers may be important, if it can be done effectively (unfortunately, I expect it is too transient to help much). So, overall, looking at the data in Fig. 5 I am not sure I see convincing evidence of significant differences in progress past stage 8. Fig. 5d is the most convincing and perhaps summing all post-stage 8 stages shows significant differences. The authors know their data better than I do, so perhaps they can explain why my arguments here are incorrect. (If my arguments are valid and Fig. 5d is sufficient, then I think it is important for the authors to explain that accumulation of stage 14 egg chambers has two contributing factors). If not, then one possible solution may be available through the Oct-b2R mutation. Does this mutation also block egg-laying? If not, then stage 14 accumulation would be due only to a change in influx and not efflux (but I suspect the answer is that egg-laying is blocked and is not rescued by OA). Another possible solution is to compare the mutant strains with and without good nutrition because the lack of efflux from stage 14 should be the same in both cases. Perhaps the authors already conducted that experiment. Some relevant data are in Fig. 6f. I am suggesting that the mutants in rich medium are used in the first evaluations of whether they affect progress past stage 8.

We are afraid that this 2nd concern is invalid because we clearly addressed the exact concern in our manuscript, and thus chose the time points to avoid that specific egg-laying issue. We are afraid that the reviewer has overlooked the time points and our explanation regarding the time points.

Namely, we chose, 1 *dpe*, 1.5 *dpe* and 2 *dpe* to avoid any confusion caused by the egg-laying defects; under our protein-starvation conditions, flies do not lay eggs until 2 *dpe*.

It was written in the original manuscript:

“We examined the ovaries of virgins from immediately after they eclosed, in wild-type (Canton-S or *w¹¹¹⁸*), *tbh^{nM18}* (hereafter called *tbh^{-/-}*) (a null allele that does not produce OA)⁵⁰, and *tbh^{+/nM18}* flies (hereafter called *tbh^{+/-}*) at 0-day post-eclosion (*dpe*), 1 *dpe*, 1.5 *dpe*, and 2 *dpe* upon protein starvation. We picked those time points because flies do not lay eggs until 2 *dpe*.”

Furthermore, we repeatedly showed our point as below;

1. **Fig. 5e**, when flies are under protein-starvations; Even at 1.5 *dpe* at which time point flies rarely lay eggs even under well-fed condition, *Tbh^{nM18}* mutants produce average 7.3 ± 5 (n=17), whereas *Tbh^{+/nM18}* control does not produce any stage 14 egg chambers.

2. **Fig. 6d**, even under complete starvation condition (with no nutrients at all), at 2 *dpe*, *Tbh^{nM18}* mutants produce 6.7 ± 4.8 of stage 14 egg chambers (n=99), while two controls, Canton-S and *w¹¹¹⁸* do not (0 and 0.5, respectively).

3. **Fig 6b**, the numbers of stage 14 egg chambers are similar in *Tbh* and *octβ2R* mutants, even though unlike *Tbh* mutants, *octβ2R* mutants could lay eggs. As for a reference, it is identified that OAMB is the major receptor of OA for egg laying as we wrote in the manuscript: "In *D. melanogaster*, there are four G-protein coupled OA receptors. OAMB is similar to vertebrate α-adrenergic receptors and essential for egg-laying³⁰."

All these data clearly demonstrate that accumulation of stage 14 in *Tbh* or *octβ2R* mutants is NOT due to a potential efflux problem.

REVIEWERS' COMMENTS

Reviewer #2 (Remarks to the Author):

The clarification of quiescence and that OA is not necessarily regulated to serve its roles in arrest seem fine to me.

I don't believe that stage 14 oocytes accumulate unless their further progress is halted, whether this is recognized as lack of ovulation after 2d of age or not previously recognized in 0-2d adults. I think this is seen in starved OA-deficient mutants but not in WT flies, starved or fed. What is seen in well-fed OA-deficient mutants? From that result one can deduce whether the accumulation of st. 14 oocytes is due to lack of OA alone or in conjunction with starvation. Either way, it seems to me that accumulation of st. 14 means there is not only progression beyond st. 8 but also arrest or delay at st. 14 and that both phenotypes may be due to lack of OA.

The authors can choose to incorporate the above logic or not.

In re-reading I noticed a couple of corrections to make:

line 90 "...period of short days..." ?

In Fig 6 (& perhaps others with similar data) the legend does not state that it is the number of egg chambers of a given stage per (fly/ovary)?

The paper is ready to be published.

Point-by-point response to the reviewer's comments

Dear Editor,

We addressed the reviewer's comments as below.

I don't believe that stage 14 oocytes accumulate unless their further progress is halted, whether this is recognized as lack of ovulation after 2d of age or not previously recognized in 0-2d adults. I think this is seen in starved OA-deficient mutants but not in WT flies, starved or fed. What is seen in well-fed OA-deficient mutants? From that result one can deduce whether the accumulation of st. 14 oocytes is due to lack of OA alone or in conjunction with starvation. Either way, it seems to me that accumulation of st. 14 means there is not only progression beyond st. 8 but also arrest or delay at st. 14 and that both phenotypes may be due to lack of OA. The authors can choose to incorporate the above logic or not.

We appreciate the reviewer's deduction and argument, which we have already addressed in our manuscript. It was written:

"Under the no nutrient condition, control virgins produce no or few stage 14 egg chambers, whereas *tbh*^{-/-} virgins produce stage 14 egg chambers at a **similar rate** to that of **Canton-S virgins fed** on CSY standard media (Fig. 6d. Compare *tbh*^{-/-} in Fig. 6d to 2.5% in Fig. 6f).

That is, **fed wild type flies** contain **the similar number of S14 eggs** seen in **starved *tbh*^{-/-} virgins**, indicating that 15-20 of S14 egg chambers are within a normal range of S14 egg numbers that fed wild type virgins contain.

We also provide three references regarding that virgin flies delay laying unfertilized eggs to support our observation regarding the number of S14 seen in fed 2 *dpe* wild type virgins.

(1) Wyman, R. reported in his article, "The temporal stability of the *Drosophila* oocyte. *J. Embryol. exp. Morph.* **50**, 131-144 (1979)" that "Retention of eggs by virgin females: When held isolated for long periods in yeasted dishes, virgin females eventually lay unfertilized eggs. Fifty virgin females were collected upon emergence and placed individually in yeasted dishes. They laid their first eggs (unfertilized) between day 3 and day 11 (on page 139)."

(2) Spradling, A. C. wrote in the book chapter, "Developmental genetics of oogenesis. Development of *Drosophila melanogaster*. Cold Spring Harbor, NY: Cold Spring Harbor Press, **1**, 1-70 (1993) that "*Drosophila* females are loath to deposit even mature eggs unless relative humidity, presence of nutrients, and seemingly idiosyncratic factors signal the presence of an environment hospitable for larval development. Even eggs held for 1 week or more after maturity usually develop normally once the female is provided with the missing conditions (on page 7)."

(3) Greenblatt *et al* (Greenblatt, E. J., Obniski, R., Mical, G. & Spradling, A. C. Prolonged ovarian storage of mature *Drosophila* oocytes dramatically increases meiotic spindle instability. *eLife* **8**, e49455, 2019) wrote that "Newly eclosed virgin female flies with immature ovaries are fed a nutrient-rich yeast paste that stimulates exactly two young follicles per ovariole to develop to maturity past a nutrient-sensitive checkpoint at stage 8. Withdrawal of the yeast food after 24 hr prevents any additional follicles from passing the checkpoint, however oocyte and maternal physiology are not adversely affected. In the absence of mating, the mature eggs are stored in the ovary indefinitely and not replaced (on page 2)."

Because we failed to find any scientific basis to support the reviewer's statement that "I think this is seen in starved OA-deficient mutants but not in WT flies, starved or **fed**." – since a fed fly CAN contain that many S14 eggs –, and because the concern is rather contradictory to our data presented, our own observations, and the literature, we did not make any amendment regarding this comment. We appreciate the reviewer's understanding.

line 90 "...period of short days..."

The revised sentence is now read:

Oocytes in *D. melanogaster* remain quiescent under short photoperiod conditions at low temperature or starvation.

In Fig 6 (& perhaps others with similar data) the legend does not state that it is the number of egg chambers of a given stage per (fly/ovary)?

We added:

b-f Numbers of egg chambers per fly were shown.